# Exploring Neural Network Representational Similarity using Filter Subspaces

## Abstract

Analyzing representational similarity in neural networks is crucial to numerous tasks, such as interpreting or transferring deep models. One typical approach is to input probing data into convolutional neural networks (CNNs) as stimuli to reveal their deep representation for model similarity analysis. Those methods are often computationally expensive and stimulus-dependent. By representing filter subspace in a CNN as a set of filter atoms, previous work has reported competitive performance in continual learning by learning a different set of filter atoms for each task while sharing common atom coefficients across tasks. Inspired by this observation, in this paper, we propose a new paradigm for reducing representational similarity analysis in CNNs to filter subspace distance assessment. Specifically, when filter atom coefficients are shared across networks, model representational similarity can be significantly simplified as calculating the cosine distance among respective filter atoms, to achieve *millions of times* computation reduction. We provide both theoretical and empirical evidence that this simplified filter subspace-based similarity preserves a strong linear correlation with other popular stimulus-based metrics, while being significantly more efficient and robust to probing data. We further validate the effectiveness of the proposed method in various applications, such as analyzing training dynamics as well as in federated and continual learning. We hope our findings can help further explorations of real-time large-scale representational similarity analysis in neural networks.

## 1 Introduction

Deep neural networks have shown unprecedented performance in a large variety of tasks (Krizhevsky et al., 2012; Ronneberger et al., 2015). The cornerstone to the success is the deep representation learned by neural networks (NNs), which contains high-level semantic information about a task. By viewing deep representation as to the characterization of each task in a high-dimensional space, the representational similarity between a pair of deep models can be exploited to understand the intrinsic relationship between associated tasks. In this way, the representational similarity provides a way to open the black box of deep learning by showing the training dynamics (Kornblith et al., 2019), and it further empowers machine learning systems with the ability to transfer knowledge from one task to another (Huang et al., 2021a).

Previous works (Raghu et al., 2017; Morcos et al., 2018) measure representational similarity directly relying on deep representations revealed by input data. These approaches introduce heavy computation from both the forward pass of numerous stimulus inputs and the calculation of high-dimensional covariance matrices. Since these similarity metrics are stimulus-dependent, their quality can potentially deteriorate when probing data are inappropriately chosen, scarce or unavailable.

We are inspired by the continual learning framework in Miao et al. (2021), where a group of tasks is simultaneously modeled using NNs by learning for each task a different set of filter atoms while sharing common atom coefficients across tasks. Miao et al. (2021) has in detail analyzed and validated this framework in a continual learning context. In the above setting, it is easy to observe that the representation variations across different NNs now become dominated by respective filter atoms. Thus, Miao et al. (2021) adopts in experiments filter subspace distance to assess task relevancy, however, without formal justification.

In this paper, we formally explore NN representational similarity using filter subspace distance, with detailed theoretical and empirical justifications. We first simplify the filter subspace distance

to the cosine distance of two sets of filter atoms, to eliminate heavy computation of singular value decomposition in calculating principal angles. Then, we show both theoretically and empirically that the obtained filter atom-based similarity preserves a strong linear correlation with other popular stimulus-dependent similarity measures such as CCA (Raghu et al., 2017). Our representational similarity is also immune to inappropriate choices of probing data, while stimulus-dependent metrics can be perturbed drastically.

The proposed filter atom-based similarity shows extreme efficiency in both memory and computation. Since the similarity computation does not involve network forward pass, no GPU memory access is required, whereas other stimulus-based measures consume the same amount of GPU memory as regular inference. On the other hand, the proposed method involves only inner product calculations on filter atoms, which takes neglectable time for similarity evaluation. The evaluation time of stimulus-based measures includes the time of both the forward pass of probing data and the calculation of high-dimensional covariance matrices. We report later the dramatically improved evaluation time of the proposed method against other popular method, e.g., CKA (Kornblith et al., 2019).

We further validate our atom-based similarity for knowledge transfer with various continual learning and federated learning tasks. In both settings, we fix the atom coefficients, learn the filter atoms for each task, and finally conduct knowledge transfer among tasks by recalling the most similar models for the ensemble. Compared with stimulus-based similarity metrics, the proposed measure achieves competitive performance with *millions of times* reduction in the computational cost.

We summarize our contributions as follows,

- We formally explore NN representational similarity measure using filter subspace distance.
- We show both theoretically and empirically that the proposed filter atom-based measure preserves a strong linear correlation with other popular stimulus-dependent measures while being significantly more robust and efficient in both memory and computation.
- We demonstrate the effectiveness of the proposed similarity measure under various example settings, such as analyzing training dynamics as well as in federated and continual learning.

## 2 METHODOLOGY

In this section, we first provide a filter subspace formulation for NNs and propose a model similarity metric based on a simplified filter subspace distance. Then, we review stimulus-based representational similarities and show their limitations. We further demonstrate that under certain assumptions, the proposed measure shows a strong linear relationship with popular stimulus-based measures, while exhibiting dramatic improvement in computational efficiency and data robustness. These unique characteristics of the proposed measure can potentially enable real-time large-scale NN similarity assessment, e.g., helping fast knowledge transfer across a large number of models.

### 2.1 REPRESENTATIONAL SIMILARITY IN FILTER SUBSPACE

**Filter subspace.** As in Qiu et al. (2018), the convolutional filter $\mathbf{W} \in \mathbb{R}^{c' \times c \times k \times k}$ ($c'$ and $c$ are the number of input and output channels, $k$ is the kernel size) can be decomposed as $m$ filter atoms $\mathbf{D}[i] \in \mathbb{R}^{k \times k} (i = 1, ..., m)$, linearly combined by atom coefficients $\boldsymbol{\alpha} \in \mathbb{R}^{m \times c' \times c}$ as $\mathbf{W} = \boldsymbol{\alpha} \times \mathbf{D}$. The filter subspace is then expressed as $\mathcal{V} = \text{Span}\{\mathbf{D}[1], ..., \mathbf{D}[m]\}$. With this formulation, we consider a paradigm where atom coefficients are shared across different deep models while filter subspaces are model specific. This paradigm has been in detail analyzed and validated in Miao et al. (2021) and reports state-of-the-art performance in a continual learning context.

In this setting, we dive deep into the relationship between filter atoms and representations. For simplicity, let $c = c' = 1$, and the argument extends. Given an input image $X(b)$ ($b \in \mathcal{B}, \mathcal{B} \subset \mathbb{Z}^2$), define the local input norm $||\mathbf{X}||_{F,N_b} := (\sum_{b' \in N_b} \mathbf{X}(b - b')^2)^{1/2}$ and the convolution $\langle \mathbf{X}, w \rangle_{N_b} := \sum_{b' \in N_b} \mathbf{X}(b - b')w(b')$, where $N_b \subset \mathcal{B}$ is a local Euclidean grid centered at $b$. Then the decomposed convolution can be written as $\mathbf{Z}(b) = \sigma(\sum_{i=1}^{m} \boldsymbol{\alpha}_i \langle \mathbf{X}, \mathbf{D}_i \rangle_{N_b})$, where $\mathbf{D}[i]$ denotes the $i$-th atom, $\boldsymbol{\alpha}_i$ is the corresponded $i$-th coefficient.

**Proposition 1.** *Suppose $\mathbf{D}_u$ and $\mathbf{D}_v$ are two different sets of filter atoms for a convolutional layer with the common atom coefficients $\boldsymbol{\alpha}$, and the activation function $\sigma$ is non-expansive, we can upper bound the changes in the corresponding features $\mathbf{Z}_u, \mathbf{Z}_v$ with atom changes,*

$$||\mathbf{Z}_u - \mathbf{Z}_v||_F \leq (||\boldsymbol{\alpha}||_F \lambda) \sqrt{|\mathcal{B}|} \cdot ||\mathbf{D}_u - \mathbf{D}_v||_F, \quad with \ \lambda = \sup_{b \in \mathcal{B}} ||\mathbf{X}||_{F,N_b}, \quad (1)$$

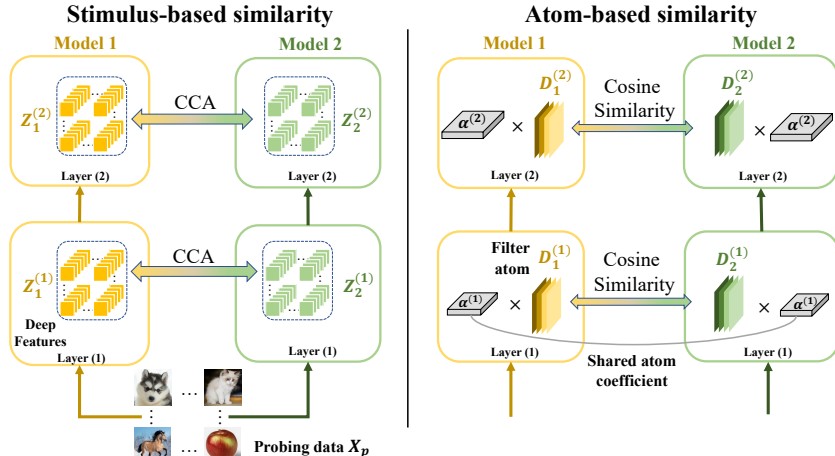

Figure 1: Comparison between our method and stimulus-based methods. (left) Stimulus-based similarity metrics, e.g., CCA, rely on probing data, and calculate the correlation between large groups of features generated by the forward pass of probing data through NNs. (right) In comparison, our atom-based method decomposes convolutional filters $\mathbf{W}$ as filter atoms $\mathbf{D}$ and atom coefficients $\boldsymbol{\alpha}$, $\mathbf{W} = \boldsymbol{\alpha} \times \mathbf{D}$, and only calculates the atom-based similarity between a small portion of parameters, *i.e.*, atoms, which is stimuli independent and computation efficient. The proposed atom-based method can achieve *millions of times* computation reduction than popular stimulus-based methods.

The proof is provided in Appendix A.1. We further empirically validate this relationship in Section 3.1.

**Filter subspace similarity**    The above theorem suggests the possibility to measure the representational similarity of two CNNs by simply measuring the distance of their filter subspaces. As proposed in Miao et al. (2021), the representational similarity of two models with different filter subspaces $\mathcal{V}_u, \mathcal{V}_v$ can be assessed by the similarity based on Grassmann distance between $\mathcal{V}_u, \mathcal{V}_v$ as,

$$\mathcal{S}_{Gras}(\mathcal{F}_u, \mathcal{F}_v) = d(\mathcal{V}_u, \mathcal{V}_v) = \frac{1}{m} \sum_i \cos\theta_i, \tag{2}$$

where $\theta_i$ is the $i$-th principal angle between $\mathcal{V}_u$ and $\mathcal{V}_v$.

However, the above metric requires costly singular value decomposition. Note that filter atoms in different models are intrinsically aligned under shared atom coefficients, which allows us to approximate the filter subspace similarity using the cosine similarity of the corresponding filter atoms. To this end, as shown in Figure 1, we propose a significantly simplified representational similarity measure with filter atom similarity.

**Definition 1.** *Suppose two convolution neural networks $\mathcal{F}_u, \mathcal{F}_v$ share atom coefficients layer-wise* *which assume to be full-rank matrices, and their model-specific filter atoms are $\mathbf{D}_u, \mathbf{D}_v$, then the atom-based representational similarity is defined as,*

$$\mathcal{S}_{Atom}(\mathcal{F}_u, \mathcal{F}_v) = \cos(\mathbf{D}_u, \mathbf{D}_v) = \frac{< vec(\mathbf{D}_u), vec(\mathbf{D}_v) >}{||vec(\mathbf{D}_u)||_F \cdot ||vec(\mathbf{D}_v)||_F}. \tag{3}$$

The above definition is a layer-wise similarity, allowing us to compare the similarity of different networks per layer, and we simply average layer-wise similarities for the network-wise similarity.

We further show that $\mathcal{S}_{Atom}$ and $\mathcal{S}_{Gras}$ are equivalent under certain assumption.

**Proposition 2.** *Assume $\mathbf{D}_u, \mathbf{D}_v \in \mathbb{R}^{k^2 \times m}$ are orthogonal matrices, then $\mathcal{S}_{Gras} = \mathcal{S}_{Atom}$.*

The proof is provided in Appendix A.1. We empirically show in Figure 2(a) that the atom-based similarity has still a strong linear correlation with the Grassmann subspace similarity even without imposing the above orthogonality over atoms.

Note that our atom-based similarity measure only involves linear operations of vectorized atoms of around 100 dimensions, which requires neglectable computation. Additionally, the proposed method depends solely on models themselves and eliminates the reliance on probing data, equipping our similarity with robustness to inappropriate choice of probing data.

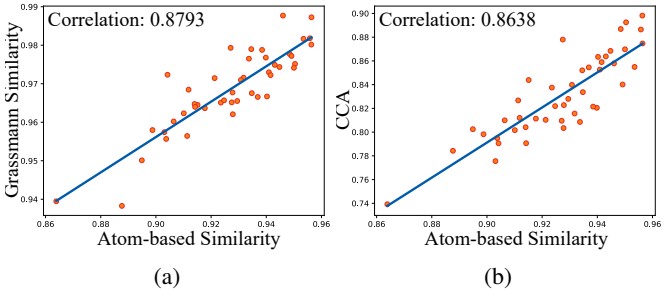

(a)                    (b)

Figure 2: (a) Correlation between Grassmann similarity and atom-based similarity; (b) Correlation between CCA and atom-based similarity. (Table) Correlation between atom-based similarity and other approaches.

## 2.2 REPRESENTATIONAL SIMILARITY IN FEATURE SPACE

Intuitively, the representational similarity can be directly assessed via features generated from different neural networks. As shown in Figure 1, it usually includes three steps to evaluate stimulus-based representational similarity between two NNs $\mathcal{F}_u$ and $\mathcal{F}_v$: (1) Collect an appropriate and sufficient amount of probing data $\mathbf{X}_p \in \mathbb{R}^{n \times c' \times h' \times w'}$ that can represent the whole data distribution, (2) Generate the feature $\mathbf{Z}_u$ and $\mathbf{Z}_v$ ($\mathbf{Z}_u, \mathbf{Z}_v \in \mathbb{R}^{n \times c \times h \times w}$) by the forward pass of probing data through different neural networks, $\mathbf{Z}_u = \mathcal{F}_u(\mathbf{X}_p, \theta_u)$ and $\mathbf{Z}_v = \mathcal{F}_v(\mathbf{X}_p, \theta_v)$, where $\theta_u, \theta_v$ denote parameters of two NNs; (3) Choose a stimulus-based metric to assess the model similarity. Several popular methods can be adopted in step (3), below we will give a brief introduction.

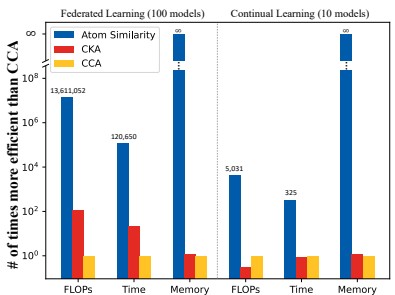

Figure 3: The ratio of the computational cost savings of our atom-based similarity over stimulus-based similarities.

**CCA.** Raghu et al. (2017) proposes to analyze the representational similarity by conducting canonical correlation analysis on two $\mathbf{Z}_u, \mathbf{Z}_v$, which is a recursive process of finding projection directions for two matrices that their correlation is maximized. Specifically, let $Q_u, Q_v$ denote the orthonormal bases of $\mathbf{Z}_u, \mathbf{Z}_v$, the CCA can be denoted as,

$$\mathcal{S}_{CCA}(\mathcal{F}_u, \mathcal{F}_v) = \sqrt{\frac{1}{c} \sum_{l=1}^{c} \sigma_l^2}, \tag{4}$$

where $\sigma_l$ denotes the $l$-th eigenvalue of $\Lambda_{u,v} = Q_u^\mathsf{T} Q_v$.

**CKA.** Kornblith et al. (2019) proposes another way to assess the similarity based on Centered Kernel Alignment (CKA). Let $K_u = \mathbf{Z}_u \mathbf{Z}_u^\mathsf{T}, K_v = \mathbf{Z}_v \mathbf{Z}_v^\mathsf{T}$ denote the Gram matrices of two feature space, the CKA is computed by,

$$\mathcal{S}_{CKA}(\mathcal{F}_u, \mathcal{F}_v) = \frac{\mathrm{HSIC}(K_u, K_v)}{\sqrt{\mathrm{HSIC}(K_u, K_v)\,\mathrm{HSIC}(K_u, K_v)}}, \tag{5}$$

where HSIC is the Hilbert-Schmidt Independence Criterion (Gretton et al., 2005).

However, in addition to the forward pass, all the aforementioned approaches further introduce significant computational costs while performing evaluation in the representation space. Nevertheless, their qualities rely heavily on the mindful choice of probing data $\mathbf{X}_p$, which undermines their robustness.

## 2.3 ALGORITHM COMPLEXITY ANALYSIS

Here, we provide a detailed comparison of computation complexity between the proposed atom-based similarity and stimulus-based similarities. Consider one convolutional layer with filter $\mathbf{W} \in \mathbb{R}^{c' \times c \times k \times k}$ ($\mathbf{W} = \boldsymbol{\alpha} \times \mathbf{D}, \mathbf{D} \in \mathbb{R}^{m \times k \times k}$) which transforms the input $\mathbf{X}_p \in \mathbb{R}^{n \times c' \times h' \times w'}$ to output $\mathbf{Z} \in \mathbb{R}^{n \times c \times h \times w}$. The complexity of our method is dominated by inner product of two tiny filter atoms, $\mathcal{O}(m \cdot k^2)$, *e.g.*, $m = 9, k = 3$ in a typical setting.

In contrast, stimulus-based similarity measure first forward feed $n$ probing samples with a complexity of $\mathcal{O}(n \cdot h'w' \cdot k^2 \cdot cc')$, then calculates covariance matrix with complexity of $\mathcal{O}(n^2 \cdot hw \cdot c)$. In total, the time complexity of CCA is $\mathcal{O}(n \cdot h'w' \cdot k^2 \cdot cc' + n^2 \cdot hw \cdot c)$.

Our method is at least $\frac{n \cdot h'w' \cdot k^2 \cdot cc' + n^2 \cdot hw \cdot c}{m \cdot k^2}$ times more efficient than stimulus-based similarity measures. As $h \gg k$, $cc' \gg m$, the computational cost of our method is negligible. For example, with 10k probing datapoints, the CCA calculation requires $1.14 \times 10^7$ times more FLOPs than the proposed atom-based similarity.

## 2.4 RELATIONSHIP WITH STIMULUS-BASED SIMILARITIES

The proposed atom-based measure not only shows extreme efficiency, but also exhibits a linear relationship with other popular stimulus-based similarities. Here, we analyze the proposed atom-based similarity $\mathcal{S}_{Atom}$ with CCA $\mathcal{S}_{CCA}$ (Raghu et al., 2017). Suppose forward passes of decomposed convolutional layer for $\mathcal{F}_u$ and $\mathcal{F}_v$ are $\mathbf{Z}_u = \boldsymbol{\alpha}\mathbf{X}_p\mathbf{D}_u$, $\mathbf{Z}_v = \boldsymbol{\alpha}\mathbf{X}_p\mathbf{D}_v$, respectively. To start with, we show that the $\mathcal{S}_{CCA}$ is upper bounded by the proposed $\mathcal{S}_{Atom}$.

**Theorem 1.** *Let* $\mathcal{T} = \mathrm{Tr}(\mathbf{X}_p^\intercal \boldsymbol{\alpha}^\intercal \boldsymbol{\alpha}\mathbf{X}_p), \mathcal{C} = \sigma_{min}(\mathbf{X}_p^\intercal \boldsymbol{\alpha}^\intercal \boldsymbol{\alpha}\mathbf{X}_p)$. *Assume* $\mathcal{K}(\mathbf{Z}_u^\intercal \mathbf{Z}_u), \mathcal{K}(\mathbf{Z}_v^\intercal \mathbf{Z}_v) \leq \gamma$. *Then* $\mathcal{S}_{CCA}(\mathcal{F}_u, \mathcal{F}_v)$ *is upper bounded by* $\mathcal{S}_{Atom}(\mathcal{F}_u, \mathcal{F}_v)$,

$$\frac{\mathcal{C}}{\gamma c^{\frac{3}{2}} \mathcal{T}} \cdot \mathcal{S}_{CCA}(\mathcal{F}_u, \mathcal{F}_v) \leq \mathcal{S}_{Atom}(\mathcal{F}_u, \mathcal{F}_v), \tag{6}$$

where $\mathrm{Tr}(\cdot)$ denotes trace of a matrix, $\sigma_{min}$ indicates the minimum eigenvalue, $\mathcal{K}(A)$ denotes the condition number of matrix $A$. We provide the proof in Appendix A.1.

Since $\mathcal{S}_{CCA}$ is stimulus-dependent, the calculated value varies depending on the choice of probing data, and the value range shows bounded by our atom-based similarity, as in the theorem above.

With additional assumptions imposed, we can further show a simple linear relationship between CCA and our atom-based similarity.

**Assumption 1.** *Suppose the diagonal elements of* $\mathbf{Z}_u^\intercal \mathbf{Z}_u$, $\mathbf{Z}_u^\intercal \mathbf{Z}_v$ *and* $\mathbf{Z}_v^\intercal \mathbf{Z}_v$ *are larger than non-diagonal element, i.e.,* $(\mathbf{Z}_u^\intercal \mathbf{Z}_u)_{ii} \gg (\mathbf{Z}_u^\intercal \mathbf{Z}_u)_{ij}$.

The Assumption 1 suggests different channels of feature $\mathbf{Z}$ have a low correlation. Reducing channel-wise dependencies has been studied in Zhang et al. (2021) and has been shown to benefit model stability.

**Theorem 2.** *If Assumption 1 holds,* $\mathcal{S}_{CCA}(\mathcal{F}_u, \mathcal{F}_v)$ *is approximately linear to atom-based similarity,*

$$\frac{\sqrt{c}}{\gamma_1 \gamma_2 \gamma_3} \cdot \mathcal{S}_{CCA}(\mathcal{F}_u, \mathcal{F}_v) = \mathcal{S}_{Atom}(\mathcal{F}_u, \mathcal{F}_v), \tag{7}$$

where $\gamma_1$, $\gamma_2$ and $\gamma_3$ contain higher order of features, which can be found in detail with the proof in Appendix A.1.

As in Figure 2, we empirically observe the linear correlation between CCA and atom-based similarity, which agrees with our theoretical findings. In addition, we find that the proposed similarity also shows a strong correlation with CKA with different kernels.

## 3 EXPERIMENTS

In this section, we first validate our theorems with several simple experiments, and then demonstrate various applications of the proposed atom-based similarity in efficiently analyzing training dynamics as well as in federated and continual learning scenarios.

### 3.1 SIMPLE VALIDATION EXPERIMENTS

We empirically validate that the change of features is bounded by the change of atoms, and the near-linear relationship between atom-based and stimulus-based similarity. Besides, we demonstrate the limit of stimulus-based similarities, as well as the verification of our assumption and theorems.

**Representation dependency on filter atoms.** We first validate the dependency of deep features on filter atoms in Proposition 1 with a simple experiment. The model $\mathcal{F}$ here is a 2-layer CNN with coefficient $\boldsymbol{\alpha}$ and atom $\mathbf{D}$ generated from normal distribution $\mathcal{N}(0,1)$. The input sample $\mathbf{X}$ is also generated from normal distribution $\mathcal{N}(0,1)$. Figure 4(a) shows the relation between $\|\mathbf{Z}_u - \mathbf{Z}_v\|_F$ and $\|\mathbf{D}_u - \mathbf{D}_v\|_F$ by fixing coefficient $\boldsymbol{\alpha}$ and input sample $\mathbf{X}$ and randomly varying filter atoms $\mathbf{D}$.

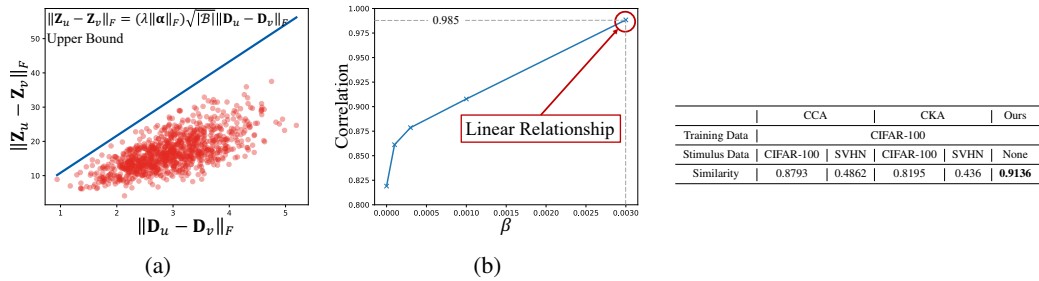

(a)          (b)

Figure 4: (a) The change of features $\|\mathbf{Z}_u - \mathbf{Z}_v\|_F$ is bounded by the change of atoms $\|\mathbf{D}_u - \mathbf{D}_v\|_F$. (b) The channel decorrelation leads to a higher correlation between CCA and atom-based similarity. And the correlation can reach 0.985 with $\beta = 3 \times 10^{-3}$, which means a near linear relation between CCA and atom-based similarity. (Table) The performance of stimulus-based similarities can be compromised by poorly selected stimulus data. For two models trained on CIFAR-100, they have high CCA and CKA similarities with stimuli from CIFAR-100 but low similarities with stimuli from SVHN. In contrast, our atom-based similarity does not depend on stimulus data and shows a high similarity between two networks as expected.

All the points are below the line which is the bound provided by Proposition 1, reflecting that the representation variations are dominated by filter atoms.

**Correlation of CCA and atom-based similarity.** We next empirically verify that CCA and atom-based similarity have a strong correlation. In this experiment, 10 tasks are generated from CIFAR100 (Krizhevsky et al., 2009) with 10 classes in each task. Only the filter atoms of each task are trained while the atom coefficients are fixed. We calculate CCA and atom-based similarity among 45 pairs of models. The correlation between CCA and atom-based similarity is *0.8638* which is shown in Figure 2(b). Similarly, the correlation between CKA and atom-based similarity is also reported in Figure 2 (Table). These results clearly show that the proposed atom-based similarity has high linear relationship with popular stimulus-based similarities, which agrees with Theorem 1 and Theorem 2.

**Effect of channel decorrelation.** We further design a regularization term $\beta \sum_{i \neq j} (\mathbf{Z}_u^\intercal \mathbf{Z}_u)_{ij}^2$ to approach $(\mathbf{Z}_u^\intercal \mathbf{Z}_u)_{ii} \gg (\mathbf{Z}_u^\intercal \mathbf{Z}_u)_{ij}$ in Assumption. 1. As shown in Figure 4(b), the correlation between CCA and atom-based similarity keeps increasing as $\beta$ increases. The correlation reaches 0.985 when $\beta = 3 \times 10^{-3}$, indicating a near-linear relationship, which is aligned with Theorem. 2.

**Limitations of stimulus-based similarities.** Depending on stimulus data, the stimulus-based similarities can be inconsistent. We expect a high value while evaluating the similarity between two models trained on the same dataset. In this experiment, we train two models on CIFAR-100. As shown in Figure 4 (Table), the CCA similarity of two models with stimuli from CIFAR-100 is 0.8793, but it drops to 0.4862 with stimuli from SVHN (Netzer et al., 2011). The CKA similarity demonstrates the same inconsistency in values with different choices of stimuli. However, our atom-based similarity of two models is 0.9136, which is aligned with our expectation.

## 3.2 LEARNING DYNAMICS

The atom-based similarity has various applications in analyzing NNs. It is capable of reflecting the data similarity and measuring the evolution of model similarity during the training time. We examine the training dynamics based on the heat map of atom-based similarities. In this experiment, AlexNet (Krizhevsky et al., 2012) is trained on CIFAR-100 (Krizhevsky et al., 2009) for 150 epochs and VGG11 (Simonyan & Zisserman, 2014) is fine-tuned on ImageNet (Russakovsky et al., 2015) for 20 epochs. For both models, we train and store atoms at each epoch. Figure 5 shows heat maps of similarities of the model among different training epochs.

Figure 5(a-c) are heat maps of the 1st, 3rd and 5th convolutional layers of Alexnet. We mark the epoch when the parameters of each layer reaches 0.99 similarity with the their states in the last epoch. The first layer reaches 0.99 similarity at epoch 36 which is earlier than final layers. In Figure 5(d-f), VGG11 shows a similar behavior. Several previous works have also indicated this bottom-up learning dynamics where layers closer to the input solidify into their final states faster than very top layers (Raghu et al., 2017; Morcos et al., 2018). Our atom-based similarity provides a highly efficient way to examine the training dynamics while showing results in accord with previous studies. Moreover, we can apply our method to calculate the similarity of a model

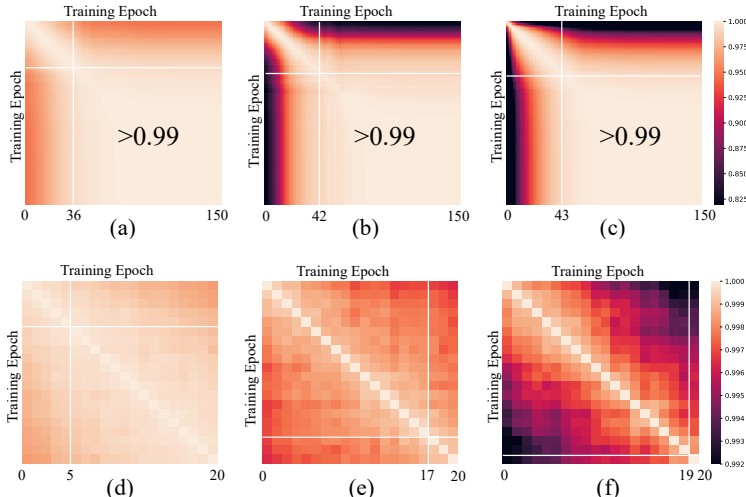

Figure 5: Layer-wise similarity matrices that show relations of model parameters of different training time points. (a)(b)(c) are the 1st, 3rd and 5th convolutional layer of AlexNet trained on CIFAR-100. (d)(e)(f) are the 1st, 4th and 8th convolutional layer of VGG11 trained on ImageNet. We mark the epoch when the parameter reaches 0.99/ 0.999 similarity to its final state with white lines. For both models, we observe bottom-up learning dynamics where layers closer to the input solidify into their final states faster than very top layers, which is in accord with previous studies (Raghu et al., 2017; Morcos et al., 2018).

trained on different tasks, so we can track the process of the same model interacting with different datasets. The details are shown in Appendix A.2.

### 3.3 FEDERATED LEARNING

Federated learning (FL) aims at learning models collaboratively by leveraging the local computational power and data of all users with the concern of privacy (McMahan et al., 2017). Personalized Federated Learning (PFL) emerges to address some challenges in FL, such as poor convergence on heterogeneous data and lack of solution personalization (Tan et al., 2022).

In this setting, our framework achieves personalization by enforcing FL models with the shared atom coefficients for all users and specific filter atoms for each user. As illustrated in Figure 7, the shared coefficients preserve the common knowledge, while user-specific atoms hold personalized information about each user. Then, we can assess model relationships with our atom-based similarity without any stimuli data, which meets the privacy requirement of the FL scenario.

The shared atom coefficients can be achieved in different ways. With our framework, the coefficient can be obtained from a model pre-trained on a public dataset or from a global model trained by other FL approaches. We can also get the coefficients by training the model locally and evolving the coefficients at each communication round.

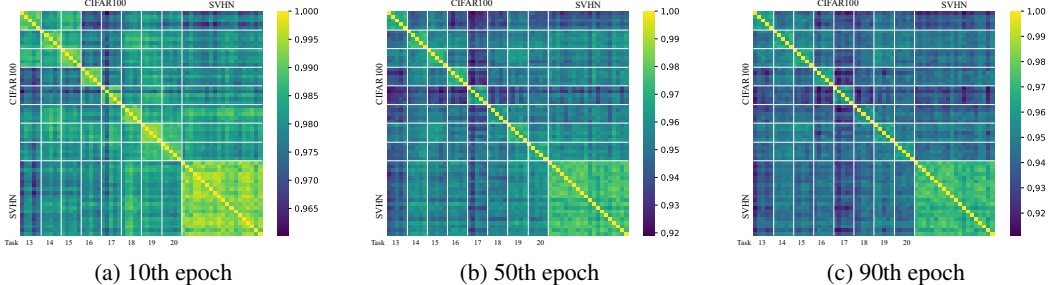

(a) 10th epoch      (b) 50th epoch      (c) 90th epoch

Figure 6: Similarity matrices that show relations among 60 users in FL with our atom-based similarity through the training process. The labels of x-axis represent the ID's of CIFAR tasks. We can clearly see user clusters in all three figures. Specifically, the last 20 clients with SVHN data show higher similarities with themselves than the first 40 clients with CIFAR data, while every five of the first 40 clients sharing the same CIFAR task also show a high similarities within themselves.

Table 1: Classification accuracy of model ensemble using different FL methods and model selection strategies: Models are selected with different similarity measures in each setting. The model ensemble performance using our atom-based method is comparable with stimulus-based methods while being millions of times faster and consuming much fewer resources.

| FL Results | Base | +Ours | +CCA (Raghu et al., 2017) | +CKA (Kornblith et al., 2019) |
|---|---|---|---|---|
| FedAvg (McMahan et al., 2017) | $83.78 \pm 0.08$ | $\mathbf{85.82 \pm 0.35}$ | $85.65 \pm 0.21$ | $85.29 \pm 0.18$ |
| Ditto (Li et al., 2021) | $82.98 \pm 0.13$ | $85.49 \pm 0.21$ | $\mathbf{85.54 \pm 0.19}$ | $85.37 \pm 0.2$ |
| FedRep (Collins et al., 2021) | $76.44 \pm 0.06$ | $\mathbf{78.35 \pm 0.24}$ | $78.18 \pm 0.18$ | $77.73 \pm 0.19$ |
| FedProx (Li et al., 2020b) | $80.6 \pm 0.1$ | $\mathbf{82.95 \pm 0.16}$ | $82.55 \pm 0.19$ | $82.86 \pm 0.16$ |
| FedPer (Arivazhagan et al., 2019) | $83.57 \pm 0.07$ | $\mathbf{85.21 \pm 0.2}$ | $84.91 \pm 0.18$ | $84.9 \pm 0.14$ |
| Pretrain | $81.77 \pm 0.08$ | $85.41 \pm 0.19$ | $85.24 \pm 0.13$ | $\mathbf{86.33 \pm 0.14}$ |
| *Similarity Computation Cost* | | | | |
| GFLOPs | | $\mathbf{0.019}$ | 258,610 | 2,225 |
| Time (s) | | $\mathbf{0.016}$ | 1930.4 | 92.6 |
| GPU Memory (MB) | | $\mathbf{0}$ | 4915 | 3965 |

Table 2: Continual Learning Results. The model ensemble using our atom-based similarity shows better result than stimulus-based methods. Our similarity is also faster and consuming much fewer resources.

| Method | CIFAR-100 | Similarity Computation Cost | | |
|---|---|---|---|---|
| | | MFLOPs | Time (s) | GPU Memory (MB) |
| AtomCL (base) | $78.11 \pm 0.13$ | - | - | - |
| +CCA Raghu et al. (2017) | $79.83 \pm 0.04$ | 35.2 | 0.26 | 1996 |
| +CKA Kornblith et al. (2019) | $80.01 \pm 0.06$ | 111 | 0.3 | 1637 |
| +Ours | $\mathbf{80.19 \pm 0.09}$ | $\mathbf{0.007}$ | $\mathbf{0.0008}$ | $\mathbf{0}$ |

**Measuring user similarity.** With the shared atom coefficients and user-specific filter atoms, we can simply get relations of users by calculating atom-based similarity. To be specific, we expect that users with similar data have a higher similarity. In this experiment, we combine two datasets, CIFAR-100 (Krizhevsky et al., 2009) and SVHN (Netzer et al., 2011), and separate data on 120 clients. Specifically, SVHN are randomly split into 20 tasks and each task contains 5 classes; CIFAR-100 are also split in the order of the class into 20 tasks and each task contains 5 classes. 20 clients are trained on 20 SVHN tasks, with each learning one task, while the other 100 clients are trained on 20 CIFAR-100 tasks, where every 5 clients equally share the data of one task by partitioning the data in a class-balanced way. The models share the same random initialization and filter atoms are trained independently without communication with other clients. The experimental details are described in Appendix A.2.

Figure 6 shows the atom-based similarity among last 40 clients of CIFAR-100 task and 20 clients of SVHN task. In Figure 6(c), where clients are trained for 90 epochs, we can clearly see the cluster of 20 SVHN clients: The SVHN clients have a higher similarity among themselves but are dissimilar from CIFAR clients. Every 5 CIFAR clients who share the same task also have a high similarity among themselves. We can see the cluster appears at the early stage of training in Figure 6(a)(b). It can be applied to quickly find clusters in FL (Tan et al., 2022). The full results of all 120 clients are shown in Appendix Figure 8. We compare atom-based similarity with CCA and CKA. Note that stimulus-based similarities need probing data which could violate the privacy requirement of FL.

The computational cost of three different approaches is shown in Figure 3. Notably, calculating the atom-based similarity is significantly faster (*million* times), requiring *0* GPU memory usage than stimuli-based methods. Note that the advantages in computational efficiency of atom-based similarity become more prominent as the number of models increases.

**Improving personalized model with ensemble of similar users.** Once we get the relationships of users, we can further improve the accuracy of the current model by the ensemble of similar models, which is effective to mitigate the data heterogeneity problem in FL. The experiment is described in detail in Appendix A.2. The final results are shown in Table 1. With ensemble, the accuracies of all FL methods can be improved. Note that the results of model ensemble selected by our atom-based similarity are comparable with stimulus-based methods while consuming much fewer resources.

### 3.4 CONTINUAL LEARNING

Continual learning is an open problem in machine learning in which data from multiple tasks arrive sequentially and the model is learned to adapt to new tasks while not forgetting the knowledge from

the past (Parisi et al., 2019). Note that some of the tasks in continual learning are related, so models trained with these tasks can be benefited from aggregating knowledge from each other. We adopt the setting in Miao et al. (2021), and apply atom-based similarity to find related models. Specifically, we *10-Split* CIFAR-100 dataset, where the 100 classes is broken down into 10 tasks with 10 classes per task. We train AlexNet including atoms and atom coefficients on the first task, and train only the atoms on the following tasks. Then, we calculate the task similarity with atom-based similarity, and report the model ensemble result with most similar members. The accuracy and the similarity computation costs are shown in Table 2. Our method provides higher results and has faster speed compared with stimulus-based methods.

## 4 RELATED WORK

### 4.1 MODEL SIMILARITY

Representational similarity analysis (RSA) (Kriegeskorte et al., 2008) demonstrates the method of understanding brain activities by computing similarities between brain responses in different regions. Measuring the similarity of models is beneficial for understanding neural network (NN) architectures and learning dynamics (Raghu et al., 2017; Kornblith et al., 2019; Morcos et al., 2018; Dwivedi & Roig, 2019). Model similarity can be used to understand or incorporate various machine learning paradigms across different areas, including contrastive learning (Islam et al., 2021; Hua et al., 2021), knowledge distillation (Stanton et al., 2021), meta-learning (Raghu et al., 2019a), and transfer learning (Raghu et al., 2019b; Neyshabur et al., 2020; Bolya et al., 2021).

Multiple approaches are proposed to estimate the representational similarity of NNs. Some early works show that individual neurons can capture meaningful information (Bau et al., 2017; Zeiler & Fergus, 2014; Zhou et al., 2016; Bau et al., 2018). Later, gradient-based methods emerge to provide a visual explanation of deep neural networks (Selvaraju et al., 2017). Current popular representational similarity methods rely on features of NN. Raghu et al. (2017) proposes SVCCA to measure similarity by calculating the covariance matrix of the features of each layer after channel alignments. Kornblith et al. (2019) discusses the invariance properties of similarity indices and proposes CKA with consistent correspondences between layers. Stimulus-based similarities are data-dependent and computationally expensive. But our method measures the representational similarity only via atoms, a portion of model parameters, which is data-agnostic and much more efficient.

### 4.2 LEARNING PARADIGM WITH NUMEROUS MODELS

Some machine learning tasks involve numerous models. For example, in Federated learning (Tan et al., 2022), thousands of models are trained across clients. In Continual learning, there are multiple models generated across time (Kirkpatrick et al., 2017). Federated learning (FL) aims to improve the performance of the system by continuously training and aggregating models from users without collecting data (McMahan et al., 2017; Smith et al., 2017; Konečnỳ et al., 2016). FL requires communication efficiency while thousands or even millions of clients may be involved (Li et al., 2020a). It also required to achieve personalization (Tan et al., 2022; Huang et al., 2021b) considering data heterogeneity of different users (Li et al., 2020a; Kairouz et al., 2021). Estimating user similarity can effectively address these challenges in FL. Continual learning (CL) aims at providing long-term knowledge accumulation, and the main challenge is to avoid catastrophic forgetting by learning new tasks while remembering the old ones (Kirkpatrick et al., 2017; Aljundi et al., 2018; Lee et al., 2017; Zenke et al., 2017; Kolouri et al., 2019). One promising way is to store neural networks for each task (Lopez-Paz & Ranzato, 2017; Rusu et al., 2016; Yoon et al., 2018; Jerfel et al., 2019; Li et al., 2019). As the number of tasks increases, a large number of models are generated and stored. It is important to find a way to access their relations to reuse models.

## 5 CONCLUSION

In this paper, we proposed a new paradigm for reducing representational similarity analysis in CNNs to filter subspace distance assessment. We provided both theoretical and empirical evidence that the proposed filter subspace-based similarity exhibits a strong linear correlation with popular stimulus-based metrics, while being significantly more efficient and robust in probing data. It was evaluated on both federated learning and continual learning tasks, and achieves competitive performance with millions of times reduction in computational cost.

Our method currently assumes respective layers among compared CNNs to have coefficients with the same dimension. For our future work, we will explore the way to share coefficient among layers to achieve atom-based similarity with different dimensions.

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

# A APPENDIX

## A.1 THEORETICAL PROOFS

**Proposition 1.** *Suppose $\mathbf{D}_u$ and $\mathbf{D}_v$ are two different sets of filter atoms for a convolutional layer with the common atom coefficients $\boldsymbol{\alpha}$, and the activation function $\sigma$ is non-expansive, we can upper bound the changes in the corresponding features $\mathbf{Z}_u, \mathbf{Z}_v$ with atom changes,*

$$||\mathbf{Z}_u - \mathbf{Z}_v||_F \leq (||\boldsymbol{\alpha}||_F \lambda)\sqrt{|\mathcal{B}|} \cdot ||(\mathbf{D}_u - \mathbf{D}_v)||_F, \quad with \ \lambda = \sup_{b \in \mathcal{B}} ||\mathbf{X}||_{F, N_b}, \tag{8}$$

*Proof.* Recall the decomposed convolution can be expressed as,

$$\mathbf{Z} = \sigma\left(\sum_{i=1}^{m} \boldsymbol{\alpha}_i \langle \mathbf{X}, \mathbf{D}[i] \rangle_{N_b}\right) \tag{9}$$

Since $\sigma$ is non-expansive, $\forall b$ we have,

$$
\begin{aligned}
|\mathbf{Z}_u(b) - \mathbf{Z}_v(b)| &\leq |\sum_{i=1}^{m} \boldsymbol{\alpha}_i \langle \mathbf{X}, \mathbf{D}_u[i] \rangle_{N_b} - \sum_{i=1}^{m} \boldsymbol{\alpha}_i \langle \mathbf{X}, \mathbf{D}_v[i] \rangle_{N_b}| \\
&\leq ||\boldsymbol{\alpha}||_F \left(\sum_{i=1}^{m} |\langle \mathbf{X}, (\mathbf{D}_u[i] - \mathbf{D}_v[i]) \rangle_{N_b}|^2\right)^{1/2}.
\end{aligned}
\tag{10}
$$

By Cauchy-Schwarz inequality,

$$
\begin{aligned}
|\langle \mathbf{X}, (\mathbf{D}_u[i] - \mathbf{D}_v[i]) \rangle_{N_b}| &\leq ||\mathbf{X}||_{F, N_b} \cdot ||\mathbf{D}_u[i] - \mathbf{D}_v[i]||_{F, N_b} \\
&\leq \lambda \cdot ||\mathbf{D}_u[i] - \mathbf{D}_v[i]||_{F, N_b}
\end{aligned}
\tag{11}
$$

we have that

$$
\begin{aligned}
\sum_{b \in \mathcal{B}} |\mathbf{Z}_u(b) - \mathbf{Z}_v(b)|^2 &\leq ||\boldsymbol{\alpha}||_F^2 \sum_b \sum_{i=1}^{m} |\langle \mathbf{X}, (\mathbf{D}_u[i] - \mathbf{D}_v[i]) \rangle_{N_b}|^2 \\
&\leq ||\boldsymbol{\alpha}||_F^2 \sum_b \sum_{i=1}^{m} ||\mathbf{X}||_{F, N_b}^2 \cdot ||(\mathbf{D}_u[i] - \mathbf{D}_v[i])||_{F, N_b}^2 \\
&\leq (||\boldsymbol{\alpha}||_F \lambda)^2 \sum_{b,i} ||(\mathbf{D}_u[i] - \mathbf{D}_v[i])||_{F, N_b}^2
\end{aligned}
\tag{12}
$$

and observe that

$$\sum_{b,i} ||(\mathbf{D}_u[i] - \mathbf{D}_v[i])||_{F, N_b}^2 = \sum_{b \in \mathcal{B}} \sum_{i=1}^{m} ||(\mathbf{D}_u[i] - \mathbf{D}_v[i])||_{F, N_b}^2 = |\mathcal{B}| \cdot ||(\mathbf{D}_u - \mathbf{D}_v)||_F^2, \tag{13}$$

where $|\mathcal{B}|$ is the area of the domain of $\mathbf{X}$. Then Eq. 12 becomes

$$\sum_{b \in \mathcal{B}} |\mathbf{Z}_u(b) - \mathbf{Z}_v(b)|^2 \leq (||\boldsymbol{\alpha}||_F \lambda)^2 |\mathcal{B}| \cdot ||(\mathbf{D}_u - \mathbf{D}_v)||_F^2, \tag{14}$$

which proves that $||\mathbf{Z}_u - \mathbf{Z}_v||_F \leq (||\boldsymbol{\alpha}||_F \lambda)\sqrt{|\mathcal{B}|} \cdot ||(\mathbf{D}_u - \mathbf{D}_v)||_F$ as claimed.

$\square$

**Proposition 2.** *Assume filter atoms $\mathbf{D}_u, \mathbf{D}_v$ are orthogonal matrices, then $\mathcal{S}_{Gras} = \mathcal{S}_{Atom}$.*

*Proof.* Since $\mathbf{D}_u, \mathbf{D}_v \in \mathbb{R}^{k^2 \times m}$ are orthogonal matrices, i.e., $\mathbf{D}_u^T \mathbf{D}_u = \mathbf{D}_v^T \mathbf{D}_v = I$, the Grassmann similarity can be represented as,

$$\mathcal{S}_{Gras}(\mathcal{F}_u, \mathcal{F}_v) = \frac{1}{m} \sum_i^m \cos\theta_i = \frac{1}{m} \sum_i^m \sigma_i, \tag{15}$$

where $\sigma_i = \Sigma_{ii}, U\Sigma V = \mathbf{D}_u^T \mathbf{D}_v$.

$\mathcal{S}_{Atom}$ is defined as,

$$\mathcal{S}_{Atom}(\mathcal{F}_u, \mathcal{F}_v) = \cos(\mathbf{D}_u, \mathbf{D}_v) = \frac{< vec(\mathbf{D}_u), vec(\mathbf{D}_v) >}{||vec(\mathbf{D}_u)||_F \cdot ||vec(\mathbf{D}_v)||_F}. \tag{16}$$

Analyze each part separately, we have $< vec(\mathbf{D}_u), vec(\mathbf{D}_v) >= \mathrm{Tr}(\mathbf{D}_u^T \mathbf{D}_v) = \sum_i^m \sigma_i$, $||vec(\mathbf{D}_u)||_F = \sqrt{\mathrm{Tr}(\mathbf{D}_u^T \mathbf{D}_u)} = \sqrt{\mathrm{Tr}(I)} = \sqrt{m}$, and also $||vec(\mathbf{D}_v)||_F = \sqrt{m}$. In total, the atom-based similarity becomes,

$$\mathcal{S}_{Atom}(\mathcal{F}_u, \mathcal{F}_v) = \cos(\mathbf{D}_u, \mathbf{D}_v) = \frac{\sum_i^m \sigma_i}{m}, \tag{17}$$

which equals $\mathcal{S}_{Gras}$. The claimed theorem is proved.

$\square$

**Lemma 1.** *For two positive semidefinite matrices* $\mathbf{A}, \mathbf{B}$,

$$\mathrm{Tr}(\mathbf{AB}) \geq \sigma_{min}(\mathbf{A})\mathrm{Tr}(\mathbf{B}), \tag{18}$$

*where $\sigma_{min}$ denotes the minimum eigenvalue of A.*

*Proof.* It is equivalent to prove that,

$$\mathrm{Tr}((\mathbf{A} - \sigma_{min}(\mathbf{A})\mathbf{I})\mathbf{B}) \geq 0. \tag{19}$$

Let $\mathbf{C}, \mathbf{D}$ be matrices such that $\mathbf{A} - \sigma_{min}(\mathbf{A})\mathbf{I} = \mathbf{C}^\intercal \mathbf{C}, \mathbf{B} = \mathbf{D}^\intercal \mathbf{D}$, then

$$\begin{aligned}
\mathrm{Tr}((\mathbf{A} - \sigma_{min}(\mathbf{A})\mathbf{I})\mathbf{B}) &= \mathrm{Tr}(\mathbf{C}^\intercal \mathbf{C} \mathbf{D}^\intercal \mathbf{D}) \\
&= \mathrm{Tr}(\mathbf{C}\mathbf{D}^\intercal \mathbf{D}\mathbf{C}^\intercal) \\
&= \mathrm{Tr}((\mathbf{D}\mathbf{C}^\intercal)^\intercal(\mathbf{D}\mathbf{C}^\intercal)) \geq 0.
\end{aligned} \tag{20}$$

$\square$

**Theorem 1.** *Suppose the forward of decomposed convolution layer for the $u$-th model is* $\mathbf{Z}_u = \alpha \mathbf{X}\mathbf{D}_u$. *$\mathbf{Z}_u, \mathbf{Z}_v$ nearly have zero-mean since $\mathbf{X}_p$ is preprocessed to be normalized. CCA coefficient is defined as* $S(\mathbf{Z}_u, \mathbf{Z}_v) = \sqrt{\frac{1}{c}\sum_{i=1}^c \sigma_i^2}$, *where $\sigma_i^2$ denotes the $i$-th eigenvalue of $\Lambda_{u,v} = Q_u^\intercal Q_v$, $Q_u = \mathbf{Z}_u(\mathbf{Z}_u^\intercal \mathbf{Z}_u)^{-\frac{1}{2}}$. Then $\mathcal{S}(\mathbf{Z}_u, \mathbf{Z}_v)$ is upper bounded,*

$$\mathcal{S}(\mathbf{Z}_u, \mathbf{Z}_v) \leq \frac{c^{\frac{3}{2}}\mathcal{T}}{\mathcal{C}}\cos(\mathbf{D}_u, \mathbf{D}_v), \tag{21}$$

where $\mathcal{T} = \mathrm{Tr}(\mathbf{X}^\intercal \boldsymbol{\alpha}^\intercal \boldsymbol{\alpha} \mathbf{X}), \mathcal{C} = \sigma_{min}(\mathbf{X}^\intercal \boldsymbol{\alpha}^\intercal \boldsymbol{\alpha} \mathbf{X})$.

*Proof.* Consider $\mathcal{S}^2 = \frac{1}{c}\sum_{i=1}^c \sigma_i^2$.

$$\mathcal{S}^2 = \frac{1}{c}\sum_{i=1}^c \sigma_i^2 = \frac{1}{c}\mathrm{Tr}(\Lambda_{u,v}\Lambda_{u,v}^\intercal). \tag{22}$$

where

$$\mathrm{Tr}(\Lambda_{u,v}\Lambda_{u,v}^\intercal) = \mathrm{Tr}(Q_u^\intercal Q_v Q_v^\intercal Q_u) = \mathrm{Tr}(Q_v Q_v^\intercal Q_u Q_u^\intercal). \tag{23}$$

As defined above, we have

$$\begin{aligned}
Q_u Q_u^\intercal &= \mathbf{Z}_u(\mathbf{Z}_u^\intercal \mathbf{Z}_u)^{-\frac{1}{2}}(\mathbf{Z}_u^\intercal \mathbf{Z}_u)^{-\frac{1}{2}}\mathbf{Z}_u^\intercal = \mathbf{Z}_u(\mathbf{Z}_u^\intercal \mathbf{Z}_u)^{-1}\mathbf{Z}_u^\intercal \\
Q_v Q_v^\intercal &= \mathbf{Z}_v(\mathbf{Z}_v^\intercal \mathbf{Z}_v)^{-\frac{1}{2}}(\mathbf{Z}_v^\intercal \mathbf{Z}_v)^{-\frac{1}{2}}\mathbf{Z}_v^\intercal = \mathbf{Z}_v(\mathbf{Z}_v^\intercal \mathbf{Z}_v)^{-1}\mathbf{Z}_v^\intercal.
\end{aligned} \tag{24}$$

Then Equation 23 becomes,

$$\text{Tr}(\Lambda_{u,v}\Lambda_{u,v}^\mathsf{T}) = \text{Tr}(\mathbf{Z}_u(\mathbf{Z}_u^\mathsf{T}\mathbf{Z}_u)^{-1}\mathbf{Z}_u^\mathsf{T}\mathbf{Z}_v(\mathbf{Z}_v^\mathsf{T}\mathbf{Z}_v)^{-1}\mathbf{Z}_v^\mathsf{T})$$
$$= \text{Tr}((\mathbf{Z}_u^\mathsf{T}\mathbf{Z}_u)^{-1}\mathbf{Z}_u^\mathsf{T}\mathbf{Z}_v(\mathbf{Z}_v^\mathsf{T}\mathbf{Z}_v)^{-1}\mathbf{Z}_v^\mathsf{T}\mathbf{Z}_u). \tag{25}$$

By Cauchy-Schwartz Inequality,

$$\text{Tr}(\Lambda_{u,v}\Lambda_{u,v}^\mathsf{T}) \le \text{Tr}((\mathbf{Z}_u^\mathsf{T}\mathbf{Z}_u)^{-1})\,\text{Tr}((\mathbf{Z}_v^\mathsf{T}\mathbf{Z}_v)^{-1})\,\text{Tr}(\mathbf{Z}_u^\mathsf{T}\mathbf{Z}_v)^2. \tag{26}$$

Then we analyze these terms individually,

$$\text{Tr}(\mathbf{Z}_u^\mathsf{T}\mathbf{Z}_v) = \text{Tr}(\mathbf{D}_u^\mathsf{T}\mathbf{X}^\mathsf{T}\boldsymbol{\alpha}^\mathsf{T}\boldsymbol{\alpha}\mathbf{X}\mathbf{D}_v) = \text{Tr}(\mathbf{X}^\mathsf{T}\boldsymbol{\alpha}^\mathsf{T}\boldsymbol{\alpha}\mathbf{X}\mathbf{D}_v\mathbf{D}_u^\mathsf{T})$$
$$\le \text{Tr}(\mathbf{X}^\mathsf{T}\boldsymbol{\alpha}^\mathsf{T}\boldsymbol{\alpha}\mathbf{X})\,\text{Tr}(\mathbf{D}_u^\mathsf{T}\mathbf{D}_v) \le \mathcal{T} \cdot \text{Tr}(\mathbf{D}_u^\mathsf{T}\mathbf{D}_v) \tag{27}$$

As for $\text{Tr}((\mathbf{Z}_u^\mathsf{T}\mathbf{Z}_u)^{-1})$, let $\lambda_1, \lambda_2, ..., \lambda_c$ be eigenvalues for $\mathbf{Z}_u^\mathsf{T}\mathbf{Z}_u$ listed in descending order ($\lambda_1 \ge \lambda_2 \ge ... \ge \lambda_c$), and assume the condition number of $\mathbf{Z}_u^\mathsf{T}\mathbf{Z}_u$ and $\mathbf{Z}_v^\mathsf{T}\mathbf{Z}_v$ satisfy $\lambda_{max}/\lambda_{min} \le \gamma$, then,

$$\text{Tr}((\mathbf{Z}_u^\mathsf{T}\mathbf{Z}_u)^{-1}) = \sum_{i=1}^{c} \frac{1}{\lambda_i} \le c \cdot \frac{1}{\lambda_c} \le \frac{\gamma c}{\lambda_1}, \tag{28}$$

where $\lambda_1 = ||\mathbf{Z}_u^\mathsf{T}\mathbf{Z}_u||_2$, $||\cdot||_2$ denotes the operator norm induced by the vector $L_2$-norm. With the norm inequalities of any positive semidefinite matrix $A$,

$$||A||_2 \ge \frac{1}{\sqrt{c}}||A||_F \ge \frac{1}{c}||A||_* \ge \frac{1}{c}\text{Tr}(A), \tag{29}$$

where $||\cdot||_F, ||\cdot||_*$ denote the Frobenius norm and the nuclear norm, respectively.

Equation (30) then becomes,

$$\text{Tr}((\mathbf{Z}_u^\mathsf{T}\mathbf{Z}_u)^{-1}) \le c \cdot \frac{1}{||\mathbf{Z}_u^\mathsf{T}\mathbf{Z}_u||_2} \le \frac{\gamma c^2}{\text{Tr}(\mathbf{Z}_u^\mathsf{T}\mathbf{Z}_u)}. \tag{30}$$

By Lemma 1,

$$\text{Tr}(\mathbf{Z}_u^\mathsf{T}\mathbf{Z}_u) = \text{Tr}(\mathbf{D}_u^\mathsf{T}\mathbf{X}^\mathsf{T}\boldsymbol{\alpha}^\mathsf{T}\boldsymbol{\alpha}\mathbf{X}\mathbf{D}_u)$$
$$= \text{Tr}(\mathbf{X}^\mathsf{T}\boldsymbol{\alpha}^\mathsf{T}\boldsymbol{\alpha}^\mathsf{T}\mathbf{X}\mathbf{D}_u\mathbf{D}_u^\mathsf{T})$$
$$\ge \sigma_{min}(\mathbf{X}^\mathsf{T}\boldsymbol{\alpha}^\mathsf{T}\boldsymbol{\alpha}^\mathsf{T}\mathbf{X})\,\text{Tr}(\mathbf{D}_u^\mathsf{T}\mathbf{D}_u)$$
$$\ge \mathcal{C} \cdot \text{Tr}(\mathbf{D}_u^\mathsf{T}\mathbf{D}_u)$$
$$\ge \mathcal{C} \cdot ||vec(\mathbf{D}_u)||_2^2, \tag{31}$$

where $vec(\cdot)$ denotes vectorization of a matrix.

Then Equation 30 is further derived as,

$$\text{Tr}((\mathbf{Z}_u^\mathsf{T}\mathbf{Z}_u)^{-1}) \le \frac{\gamma c^2}{\mathcal{C} \cdot ||vec(\mathbf{D}_u)||_2^2}. \tag{32}$$

Similarly, we have

$$\text{Tr}((\mathbf{Z}_v^\mathsf{T}\mathbf{Z}_v)^{-1}) \le \frac{\gamma c^2}{\mathcal{C} \cdot ||vec(\mathbf{D}_v)||_2^2}. \tag{33}$$

Finally, with $\text{Tr}(\mathbf{D}_u^\mathsf{T}\mathbf{D}_v) = <vec(\mathbf{D}_u), vec(\mathbf{D}_v)>$, we have

$$\text{Tr}(\Lambda_{u,v}\Lambda_{u,v}^\mathsf{T}) \le \frac{\gamma^2\mathcal{T}^2 c^4(<vec(\mathbf{D}_u), vec(\mathbf{D}_v)>)^2}{\mathcal{C}^2||vec(\mathbf{D}_u)||_2^2 \cdot ||vec(\mathbf{D}_v)||_2^2}$$
$$\le \frac{\gamma^2\mathcal{T}^2 c^4}{\mathcal{C}^2} \cdot \cos^2(\mathbf{D}_u, \mathbf{D}_v), \tag{34}$$

and thus,

$$
\begin{aligned}
\mathcal{S}(\mathbf{Z}_u, \mathbf{Z}_v) &= \sqrt{\frac{1}{c}\operatorname{Tr}(\Lambda_{u,v}\Lambda_{u,v}^{\mathsf{T}})} \\
&\leq \frac{\gamma \mathcal{T} c^{\frac{3}{2}}}{\mathcal{C}} \cdot \mathbf{cos}(\mathbf{D}_u, \mathbf{D}_v).
\end{aligned}
\tag{35}
$$

Then the claimed theorem is proved.

$\square$

**Lemma 2.** *For two matrices* $\mathbf{A}$, $\mathbf{B}$*, their frobenius norm satisfies,*

$$
\|\mathbf{AB}\|_F = \|\mathbf{A}\|_F \|\mathbf{B}\|_F \sqrt{1 - \frac{\Delta_1}{\|\mathbf{A}\|_F^2 \|\mathbf{B}\|_F^2}},
\tag{36}
$$

*where* $\Delta_1 = \sum_{ij}(\sum_k A_{ik}^2)(\sum_k B_{kj}^2) \cdot \sin^2(\langle A_{i:}, B_{:j}\rangle)$.

*Proof.* According to the definition of frobenius norm $\|\mathbf{A}\|_F = \sqrt{\sum_{ij}|A_{ij}|^2}$ we have,

$$
\|\mathbf{AB}\|_F = \sqrt{\sum_{ij}(\sum_k A_{ik}B_{kj})^2}.
\tag{37}
$$

Note that $(\sum_i x_i y_i)^2 = (\sum_i x_i^2)(\sum_i y_i^2) \cdot \cos^2(\langle x, y \rangle) = (\sum_i x_i^2)(\sum_i y_i^2) - (\sum_i x_i^2)(\sum_i y_i^2) \cdot \sin^2(\langle x, y \rangle)$, where $\langle x, y \rangle$ is the angle of two vectors $x$ and $y$. We have,

$$
\begin{aligned}
&\sqrt{\sum_{ij}(\sum_k A_{ik}B_{kj})^2} \\
&= \sqrt{\sum_{ij}\left[(\sum_k A_{ik}^2)(\sum_k B_{kj}^2) - (\sum_k A_{ik}^2)(\sum_k B_{kj}^2) \cdot \sin^2(\langle A_{i:}, B_{:j}\rangle)\right]} \\
&= \sqrt{\sum_{ik} A_{ik}^2}\sqrt{\sum_{kj} B_{kj}^2}\sqrt{1 - \frac{\sum_{ij}(\sum_k A_{ik}^2)(\sum_k B_{kj}^2) \cdot \sin^2(\langle A_{i:}, B_{:j}\rangle)}{\sum_{ik} A_{ik}^2 \sum_{kj} B_{kj}^2}} \\
&= \|\mathbf{A}\|_F \|\mathbf{B}\|_F \sqrt{1 - \frac{\sum_{ij}(\sum_k A_{ik}^2)(\sum_k B_{kj}^2) \cdot \sin^2(\langle A_{i:}, B_{:j}\rangle)}{\|\mathbf{A}\|_F^2 \|\mathbf{B}\|_F^2}} \\
&= \|\mathbf{A}\|_F \|\mathbf{B}\|_F \sqrt{1 - \frac{\Delta_1}{\|\mathbf{A}\|_F^2 \|\mathbf{B}\|_F^2}},
\end{aligned}
\tag{38}
$$

where $A_{i:}$ is the $i$-th row of $\mathbf{A}$ and $B_{:j}$ is the $j$-th column of $\mathbf{B}$, $\Delta_1 = \sum_{ij}(\sum_k A_{ik}^2)(\sum_k B_{kj}^2) \cdot \sin^2(\langle A_{i:}, B_{:j}\rangle)$. As $A_{i:}$ and $B_{:j}$ are more correlated, $\langle A_{i:}, B_{:j}\rangle \to 0$, thus, $\Delta_1 \ll \|\mathbf{A}\|_F^2 \|\mathbf{B}\|_F^2$.

$\square$

**Lemma 3.**

$$
\|\mathbf{A}^{1/2}\|_F = \|\mathbf{A}\|_F^{1/2}(1 + \frac{\Delta_{1\mathbf{A}^{1/2}}}{\|\mathbf{A}\|_F^2})^{1/4}.
\tag{39}
$$

*Proof.* According to Lemma 2, we have,

$$
\|\mathbf{A}\|_F^2 = \|\mathbf{A}^{1/2}\|_F^4 - \Delta_1.
\tag{40}
$$

Thus,

$$
\|\mathbf{A}^{1/2}\|_F = \|\mathbf{A}\|_F^{1/2}(1 + \frac{\Delta_{1A^{1/2}}}{\|\mathbf{A}\|_F^2})^{1/4},
\tag{41}
$$

where $\Delta_{1\mathbf{A}^{1/2}} = \sum_{ij}(\sum_k (A^{1/2})^2_{ik})(\sum_k (A^{1/2})^2_{kj}) \cdot \sin^2\left(\langle (A^{1/2})_{i:}, (A^{1/2})_{:j}\rangle\right)$. As $(A^{1/2})_{i:}$ and $(A^{1/2})_{:j}$ are more correlated, $\langle (A^{1/2})_{i:}, (A^{1/2})_{:j}\rangle \to 0$, thus, $\Delta_{1A^{1/2}} \ll \|\mathbf{A}\|^2_F$.

$\square$

**Lemma 4.** *For three matrices* $\mathbf{A}$*,* $\mathbf{B}$*, and* $\mathbf{C}$*, their frobenius norm satisfies,*

$$\|\mathbf{A}\|_F = \|\mathbf{A}\|_F \|\mathbf{B}\|_F \|\mathbf{C}\|_F \sqrt{1 - \frac{\Delta_2 + \Delta_3}{\|\mathbf{A}\|^2_F \|\mathbf{B}\|^2_F \|\mathbf{C}\|^2_F}}, \tag{42}$$

*where* $\Delta_2 = \frac{1}{2}[\|\mathbf{A}\|^2_F \sum_{kj}(\sum_l B^2_{kl})(\sum_l C^2_{lj}) \cdot \sin^2\left(\langle B_{k:}, C_{:j}\rangle\right) + \|\mathbf{C}\|^2_F \sum_{il}(\sum_k A^2_{ik})(\sum_k B^2_{kl}) \cdot \sin^2\left(\langle A_{i:}, B_{:l}\rangle\right)]$ *and* $\Delta_3 = \frac{1}{2}[\sum_{ij}(\sum_k A^2_{ik})(\sum_k (BC)^2_{kj}) \cdot \sin^2\left(\langle A_{i:}, (BC)_{:j}\rangle\right) + \sum_{ij}(\sum_l (AB)^2_{il})(\sum_l C^2_{lj}) \cdot \sin^2\left(\langle (AB)_{i:}, C_{:j}\rangle\right)]$.

*Proof.* Based on Lemma 2, we have,

$$
\begin{aligned}
&\|\mathbf{ABC}\|^2_F \\
=&\|\mathbf{AB}\|^2_F \|\mathbf{C}\|^2_F - \sum_{ij}(\sum_l (AB)^2_{il})(\sum_l C^2_{lj}) \cdot \sin^2\left(\langle (AB)_{i:}, C_{:j}\rangle\right) \\
=&\|\mathbf{A}\|^2_F \|\mathbf{B}\|^2_F \|\mathbf{C}\|^2_F - \|\mathbf{C}\|^2_F \sum_{il}(\sum_k A^2_{ik})(\sum_k B^2_{kl}) \cdot \sin^2\left(\langle A_{i:}, B_{:l}\rangle\right) \\
&- \sum_{ij}(\sum_l (AB)^2_{il})(\sum_l C^2_{lj}) \cdot \sin^2\left(\langle (AB)_{i:}, C_{:j}\rangle\right)
\end{aligned}
\tag{43}
$$

Symmetrically, we also have,

$$
\begin{aligned}
&\|\mathbf{ABC}\|^2_F \\
=&\|\mathbf{A}\|^2_F \|\mathbf{BC}\|^2_F - \sum_{ij}(\sum_k A^2_{ik})(\sum_k (BC)^2_{kj}) \cdot \sin^2\left(\langle A_{i:}, (BC)_{:j}\rangle\right) \\
=&\|\mathbf{A}\|^2_F \|\mathbf{B}\|^2_F \|\mathbf{C}\|^2_F - \|\mathbf{A}\|^2_F \sum_{kj}(\sum_l B^2_{kl})(\sum_l C^2_{lj}) \cdot \sin^2\left(\langle B_{k:}, C_{:j}\rangle\right) \\
&- \sum_{ij}(\sum_k A^2_{ik})(\sum_k (BC)^2_{kj}) \cdot \sin^2\left(\langle A_{i:}, (BC)_{:j}\rangle\right)
\end{aligned}
\tag{44}
$$

Thus,

$$
\begin{aligned}
&\|\mathbf{ABC}\|^2_F \\
=&\frac{1}{2}[\|\mathbf{A}\|^2_F \|\mathbf{B}\|^2_F \|\mathbf{C}\|^2_F - \|\mathbf{A}\|^2_F \sum_{kj}(\sum_l B^2_{kl})(\sum_l C^2_{lj}) \cdot \sin^2\left(\langle B_{k:}, C_{:j}\rangle\right) \\
&- \sum_{ij}(\sum_k A^2_{ik})(\sum_k (BC)^2_{kj}) \cdot \sin^2\left(\langle A_{i:}, (BC)_{:j}\rangle\right) \\
&+ \|\mathbf{A}\|^2_F \|\mathbf{B}\|^2_F \|\mathbf{C}\|^2_F - \|\mathbf{C}\|^2_F \sum_{il}(\sum_k A^2_{ik})(\sum_k B^2_{kl}) \cdot \sin^2\left(\langle A_{i:}, B_{:l}\rangle\right) \\
&- \sum_{ij}(\sum_l (AB)^2_{il})(\sum_l C^2_{lj}) \cdot \sin^2\left(\langle (AB)_{i:}, C_{:j}\rangle\right)] \\
=&\|\mathbf{A}\|^2_F \|\mathbf{B}\|^2_F \|\mathbf{C}\|^2_F - \Delta_2 - \Delta_3,
\end{aligned}
\tag{45}
$$

where $\Delta_2 = \frac{1}{2}[\|A\|^2_F \sum_{kj}(\sum_l B^2_{kl})(\sum_l C^2_{lj}) \cdot \sin^2\left(\langle B_{k:}, C_{:j}\rangle\right) + \|C\|^2_F \sum_{il}(\sum_k A^2_{ik})(\sum_k B^2_{kl}) \cdot \sin^2\left(\langle A_{i:}, B_{:l}\rangle\right)]$ and $\Delta_3 = \frac{1}{2}[\sum_{ij}(\sum_k A^2_{ik})(\sum_k (BC)^2_{kj}) \cdot \sin^2\left(\langle A_{i:}, (BC)_{:j}\rangle\right) + \sum_{ij}(\sum_l (AB)^2_{il})(\sum_l C^2_{lj}) \cdot \sin^2\left(\langle (AB)_{i:}, C_{:j}\rangle\right)]$. Therefore,

$$\|\mathbf{ABC}\|_F = \|\mathbf{A}\|_F \|\mathbf{B}\|_F \|\mathbf{C}\|_F \sqrt{1 - \frac{\Delta_2 + \Delta_3}{\|\mathbf{A}\|^2_F \|\mathbf{B}\|^2_F \|\mathbf{C}\|^2_F}}. \tag{46}$$

As $A_{i:}$ and $B_{:l}$, $B_{k:}$ and $C_{:j}$ are more correlated, $\langle A_{i:}, B_{:l} \rangle, \langle B_{k:}, C_{:j} \rangle, \langle A_{i:}, (BC)_{:j} \rangle, \langle (AB)_{i:}, C_{:j} \rangle \to 0$, thus, $\Delta_2 \ll \|\mathbf{A}\|_F^2 \|\mathbf{B}\|_F^2 \|\mathbf{C}\|_F^2$ and $\Delta_3 \ll \|\mathbf{A}\|_F^2 \|\mathbf{B}\|_F^2 \|\mathbf{C}\|_F^2$.

$\square$

**Lemma 5.**

$$\|\mathbf{A}^{-1/2}\mathbf{B}\mathbf{C}^{-1/2}\|_F = \kappa_F(\mathbf{A}^{1/2})\kappa_F(\mathbf{C}^{1/2})\frac{\|\mathbf{B}\|_F}{\|\mathbf{A}^{1/2}\|_F\|\mathbf{C}^{1/2}\|_F}\sqrt{1 - \frac{\Delta_2 + \Delta_3}{\|\mathbf{A}^{-1/2}\|_F^2\|\mathbf{B}\|_F^2\|\mathbf{C}^{-1/2}\|_F^2}},$$
(47)

*where $\kappa_F(\mathbf{A}^{1/2})$ and $\kappa_F(\mathbf{C}^{1/2})$ are the condition number of $\mathbf{A}^{1/2}$ and $\mathbf{C}^{1/2}$, $\kappa_F(\mathbf{A}^{1/2}) = \sqrt{(\sum \sigma_i^2(\mathbf{A}^{1/2}))(\sum \frac{1}{\sigma_i^2(\mathbf{A}^{1/2})})}$ and $\kappa_F(\mathbf{C}^{1/2}) = \sqrt{(\sum \sigma_i^2(\mathbf{C}^{1/2}))(\sum \frac{1}{\sigma_i^2(\mathbf{C}^{1/2})})}$; $\sigma_i^2(\mathbf{A}^{1/2})$ are singular value of $\mathbf{A}^{1/2}$ and $\sigma_i^2(\mathbf{C}^{1/2})$ are singular value of $\mathbf{C}^{1/2}$.*

*Proof.* Based on Lemma 4, we have,

$$\|\mathbf{A}^{-1/2}\mathbf{B}\mathbf{C}^{-1/2}\|_F = \|\mathbf{A}^{-1/2}\|_F\|\mathbf{B}\|_F\|\mathbf{C}^{-1/2}\|_F\sqrt{1 - \frac{\Delta_2 + \Delta_3}{\|\mathbf{A}^{-1/2}\|_F^2\|\mathbf{B}\|_F^2\|\mathbf{C}^{-1/2}\|_F^2}}.$$
(48)

By the definition of condition number $\kappa_F(\mathbf{X}) = \|\mathbf{X}\|_F\|\mathbf{X}^{-1}\|_F = \sqrt{(\sum \sigma_i^2(\mathbf{X}))(\sum \frac{1}{\sigma_i^2(\mathbf{X})})}$,

$$\|\mathbf{A}^{-1/2}\mathbf{B}\mathbf{C}^{-1/2}\|_F = \kappa_F(\mathbf{A}^{1/2})\kappa_F(\mathbf{C}^{1/2})\frac{\|\mathbf{B}\|_F}{\|\mathbf{A}^{1/2}\|_F\|\mathbf{C}^{1/2}\|_F}\sqrt{1 - \frac{\Delta_2 + \Delta_3}{\|\mathbf{A}^{-1/2}\|_F^2\|\mathbf{B}\|_F^2\|\mathbf{C}^{-1/2}\|_F^2}}.$$
(49)

$\square$

**Theorem 2.** *Suppose the forward of decomposed convolution layer for the $u$-th model is $\mathbf{Z}_u = \alpha\mathbf{X}\mathbf{D}_u$, CCA coefficient be $S(\mathbf{Z}_u, \mathbf{Z}_v) = \sqrt{\frac{1}{c}\sum_{i=1}^c \sigma_i^2}$, where $\sigma_i^2$ denotes the $i$-th eigenvalue of $\Lambda_{u,v} = Q_u^\intercal Q_v$, $Q_u = \mathbf{Z}_u(\mathbf{Z}_u^\intercal\mathbf{Z}_u)^{-\frac{1}{2}}$. Then $\mathcal{S}(\mathbf{Z}_u, \mathbf{Z}_v)$ is approximately linear to atom-based similarity,*

$$\mathcal{S}(\mathbf{Z}_u, \mathbf{Z}_v) = \frac{\gamma_1\gamma_2\gamma_3}{\sqrt{c}}\cos(\mathbf{D}_u, \mathbf{D}_v),$$
(50)

*Proof.* Based on $S(\mathbf{Z}_u, \mathbf{Z}_v) = \sqrt{\frac{1}{c}\sum_{i=1}^c \sigma_i^2}$ and $\|\Lambda_{u,v}\|_F = \sqrt{\sum_{i=1}^c \sigma_i^2}$, where $\sigma_i$ are the singular value of $\Lambda_{u,v}$,

$$\mathcal{S} = \sqrt{\frac{1}{c}\sum_{i=1}^c \sigma_i^2} = \frac{1}{\sqrt{c}}\|\Lambda_{u,v}\|_F = \frac{1}{\sqrt{c}}\|(\mathbf{Z}_u^\intercal\mathbf{Z}_u)^{-\frac{1}{2}}\mathbf{Z}_u^\intercal\mathbf{Z}_v(\mathbf{Z}_v^\intercal\mathbf{Z}_v)^{-\frac{1}{2}}\|_F.$$
(51)

According to Lemma. 5, we have

$$\frac{1}{\sqrt{c}}\|(\mathbf{Z}_u^\intercal\mathbf{Z}_u)^{-\frac{1}{2}}\mathbf{Z}_u^\intercal\mathbf{Z}_v(\mathbf{Z}_v^\intercal\mathbf{Z}_v)^{-\frac{1}{2}}\|_F = \frac{\gamma_1\gamma_2}{\sqrt{c}}\frac{\|\mathbf{Z}_u^\intercal\mathbf{Z}_v\|_F}{\|(\mathbf{Z}_u^\intercal\mathbf{Z}_u)^{\frac{1}{2}}\|_F\|(\mathbf{Z}_v^\intercal\mathbf{Z}_v)^{\frac{1}{2}}\|_F},$$
(52)

where $\gamma_1 = \kappa_F((\mathbf{Z}_u^\intercal\mathbf{Z}_u)^{\frac{1}{2}}) \cdot \kappa_F((\mathbf{Z}_v^\intercal\mathbf{Z}_v)^{\frac{1}{2}})$ and $\gamma_2 = \sqrt{1 - \frac{\Delta_2 + \Delta_3}{\|(\mathbf{Z}_u^\intercal\mathbf{Z}_u)^{-1/2}\|_F^2\|\mathbf{Z}_u^\intercal\mathbf{Z}_v\|_F^2\|(\mathbf{Z}_v^\intercal\mathbf{Z}_v)^{-1/2}\|_F^2}}$.

As $\mathbf{Z}_u = \alpha\mathbf{X}\mathbf{D}_u$ and $\mathbf{Z}_v = \alpha\mathbf{X}\mathbf{D}_v$, we have

$$\begin{aligned}
&\frac{\gamma_1\gamma_2}{\sqrt{c}}\frac{\|\mathbf{Z}_u^\intercal\mathbf{Z}_v\|_F}{\|(\mathbf{Z}_u^\intercal\mathbf{Z}_u)^{\frac{1}{2}}\|_F\|(\mathbf{Z}_v^\intercal\mathbf{Z}_v)^{\frac{1}{2}}\|_F}\\
&= \frac{\gamma_1\gamma_2}{\sqrt{c}}\frac{\|\mathbf{D}_u^\intercal\mathbf{X}^\intercal\alpha^\intercal\alpha\mathbf{X}\mathbf{D}_v\|_F}{\|(\mathbf{D}_u^\intercal\mathbf{X}^\intercal\alpha^\intercal\alpha\mathbf{X}\mathbf{D}_u)^{\frac{1}{2}}\|_F\|(\mathbf{D}_v^\intercal\mathbf{X}^\intercal\alpha^\intercal\alpha\mathbf{X}\mathbf{D}_v)^{\frac{1}{2}}\|_F}.
\end{aligned}$$
(53)

According to Lemma 3,

$$
\begin{aligned}
&\frac{\gamma_1\gamma_2}{\sqrt{c}}\frac{\|\mathbf{D}_u^\intercal\mathbf{X}^\intercal\boldsymbol{\alpha}^\intercal\boldsymbol{\alpha}\mathbf{X}\mathbf{D}_v\|_F}{\|(\mathbf{D}_u^\intercal\mathbf{X}^\intercal\boldsymbol{\alpha}^\intercal\boldsymbol{\alpha}\mathbf{X}\mathbf{D}_u)^{\frac12}\|_F\|(\mathbf{D}_v^\intercal\mathbf{X}^\intercal\boldsymbol{\alpha}^\intercal\boldsymbol{\alpha}\mathbf{X}\mathbf{D}_v)^{\frac12}\|_F}\\
&=\frac{\gamma_1\gamma_2\gamma_3}{\sqrt{c}}\frac{\|\mathbf{D}_u^\intercal\mathbf{X}^\intercal\boldsymbol{\alpha}^\intercal\boldsymbol{\alpha}\mathbf{X}\mathbf{D}_v\|_F}{\|(\mathbf{D}_u^\intercal\mathbf{X}^\intercal\boldsymbol{\alpha}^\intercal\boldsymbol{\alpha}\mathbf{X}\mathbf{D}_u)\|_F^{\frac12}\|(\mathbf{D}_v^\intercal\mathbf{X}^\intercal\boldsymbol{\alpha}^\intercal\boldsymbol{\alpha}\mathbf{X}\mathbf{D}_v)\|_F^{\frac12}},
\end{aligned}
\tag{54}
$$

where $\gamma_3 = (1+\frac{\Delta_1}{\|(\mathbf{D}_u^\intercal\mathbf{X}^\intercal\boldsymbol{\alpha}^\intercal\boldsymbol{\alpha}\mathbf{X}\mathbf{D}_u)\|_F^2})^{-\frac14}(1+\frac{\Delta_1}{\|(\mathbf{D}_v^\intercal\mathbf{X}^\intercal\boldsymbol{\alpha}^\intercal\boldsymbol{\alpha}\mathbf{X}\mathbf{D}_v)\|_F^2})^{-\frac14}$.

As Assumption 1 holds, it becomes

$$
\begin{aligned}
&\frac{\gamma_1\gamma_2\gamma_3}{\sqrt{c}}\frac{\|\mathbf{D}_u^\intercal\mathbf{X}^\intercal\boldsymbol{\alpha}^\intercal\boldsymbol{\alpha}\mathbf{X}\mathbf{D}_v\|_F}{\|(\mathbf{D}_u^\intercal\mathbf{X}^\intercal\boldsymbol{\alpha}^\intercal\boldsymbol{\alpha}\mathbf{X}\mathbf{D}_u)\|_F^{\frac12}\|(\mathbf{D}_v^\intercal\mathbf{X}^\intercal\boldsymbol{\alpha}^\intercal\boldsymbol{\alpha}\mathbf{X}\mathbf{D}_v)\|_F^{\frac12}}\\
&=\frac{\gamma_1\gamma_2\gamma_3}{\sqrt{c}}\frac{\|\mathbf{D}_u^\intercal\mathbf{D}_v\|_F\|\mathbf{X}^\intercal\boldsymbol{\alpha}^\intercal\boldsymbol{\alpha}\mathbf{X}\|_F}{\|\mathbf{D}_u^\intercal\|_F^{\frac12}\|\mathbf{X}^\intercal\boldsymbol{\alpha}^\intercal\boldsymbol{\alpha}\mathbf{X}\|_F^{\frac12}\|\mathbf{D}_u\|_F^{\frac12}\|\mathbf{D}_v^\intercal\|_F^{\frac12}\|\mathbf{X}^\intercal\boldsymbol{\alpha}^\intercal\boldsymbol{\alpha}\mathbf{X}\|_F^{\frac12}\|\mathbf{D}_v\|_F^{\frac12}}\\
&=\frac{\gamma_1\gamma_2\gamma_3}{\sqrt{c}}\frac{\|\mathbf{D}_u^\intercal\mathbf{D}_v\|_F}{\|\mathbf{D}_u\|_F\|\mathbf{D}_v\|_F}\\
&=\frac{\gamma_1\gamma_2\gamma_3}{\sqrt{c}}\cos(\mathbf{D}_u,\mathbf{D}_v).
\end{aligned}
\tag{55}
$$

Thus, we have

$$
\mathcal{S}(\mathbf{Z}_u,\mathbf{Z}_v) = \frac{\gamma_1\gamma_2\gamma_3}{\sqrt{c}}\cos(\mathbf{D}_u,\mathbf{D}_v).
\tag{56}
$$

$\square$

## A.2 Experiment Settings

**Model training of Federated Learning.** In each experiment we have 100 clients in total and sample a ratio $r = 0.1$ of all the clients on every round. All models are randomly initialized and trained for $T = 100$ communication rounds for the CIFAR datasets. At each round, the client executes 15 epochs of SGD with momentum to train the local model, the learning rate is 0.01 and momentum is 0.9. Accuracies are computed by taking the average local accuracies for all users at the final communication round. As shown in the Table 3, we have different settings for CIFAR-10 and CIFAR-100. For example, $(100, 2)$ means 100 clients with 2 classes on each client. For each method, the training takes about 12 hours on Nvidia RTX A5000.

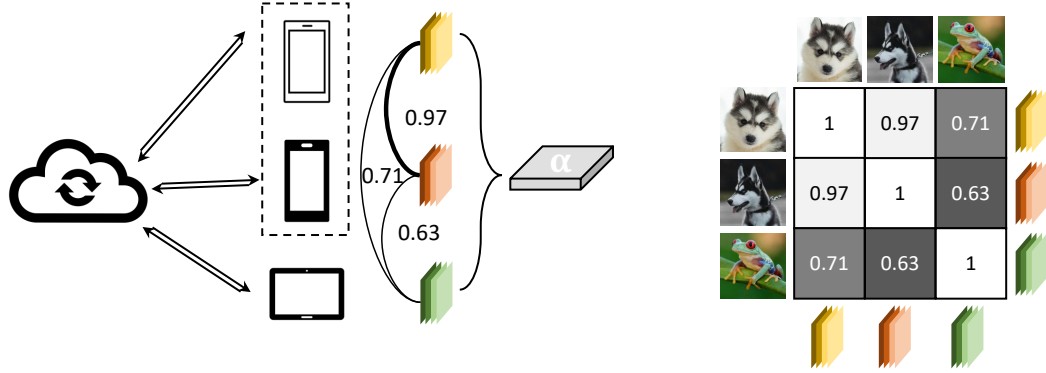

Figure 7: The shared coefficients and user-specific atoms represent common knowledge and personalized information. The atom-based similarity is used to calculate the relations among users. Users with heterogeneous data result in lower similarity, as illustrated in a similarity matrix.

Table 3: Compare accuracy with different approaches

| (# client, # classes per client) | CIFAR-100 | | CIFAR-10 | | |
|---|---|---|---|---|---|
| | (100, 5) | (100, 20) | (100, 2) | (100, 5) | (1000, 2) |
| FedAvg | 82.39 | 62.92 | 86.37 | 70.63 | 86.12 |
| FedProx | 80.77 | 59.7 | 85.90 | 69.94 | 84.83 |
| FedPer | 81.46 | 62.52 | 81.74 | 68.24 | 81.74 |
| FedRep | 72.98 | 37.71 | 80.55 | 67.3 | 82.98 |
| Local | 81.21 | 49.25 | 90.24 | 72.05 | 97.80 |
| Ours | 81.03 | 52.13 | 83.37 | 65.63 | 82.54 |

(a) 10th epoch        (b) 50th epoch        (c) 90th epoch

Figure 8: Similarity matrices that show relations among 120 users in FL with our atom-based similarity through the training process.

**Evolving shared atom coefficients.** we can get shared atom coefficients by evolving them during the communication, apart from pre-trained models or from other FL approaches. In FL, the server aggregates models from every involved client at each communication round. Our method enforces the model with a shared atom coefficient and atoms. At each communication round, the clients perform training on both atoms and atom coefficients with locally stored data. Then, the server aggregates only the atom coefficients of selected clients to get an updated coefficient. In this way, we can get a shared coefficient across clients.

**Comparison with other FL approaches.** We compare our approach by evolving shared atom coefficients with various personalized federated learning methods and federated learning methods with local finetuning. Among these methods, FedPer (Arivazhagan et al., 2019) and FedRep(Collins et al., 2021) have the similar ideas by learning shared global representation and personalized local heads. Ditto (Li et al., 2021) and FedProx (Li et al., 2020b) induce global regularization to improve the model performance. We also compare our method with FedAvg (McMahan et al., 2017). FedRep (Collins et al., 2021) approaches the common knowledge with shared representation. The codes are adapted from [1]. We evaluate the test accuracy on CIFAR-10 and CIFAR-100 with different FL setting. As shown in Table 3, our method achieves comparable performance among different methods.

**Fine-tuning models for ensemble.** We select 3 models with different similarity measures for ensemble. For feature-based similarity methods, we randomly select 1000 examples from CIFAR-100 dataset. The fully-connected layer of each model is fine-tuned on the user's local data with 100 epochs. The fine-tuning takes about 12 hours on Nvidia RTX A5000. After fine-tuning, the accuracy is measured on local test data, with the predictions of current model and 3 selected models.

**Similar representations across datasets.** Similar to (Kornblith et al., 2019), we can use atom-based similarity to compare networks trained on different datasets. In Figure 9(a), we show that pairs of models that are both trained on CIFAR-10 and CIFAR-100 have high atom-based similarities. Models learned on two datasets respectively still show high similarity. In contrast, similarities between trained and untrained models are significantly lower.

---

[1] https://github.com/lgcollins/FedRep

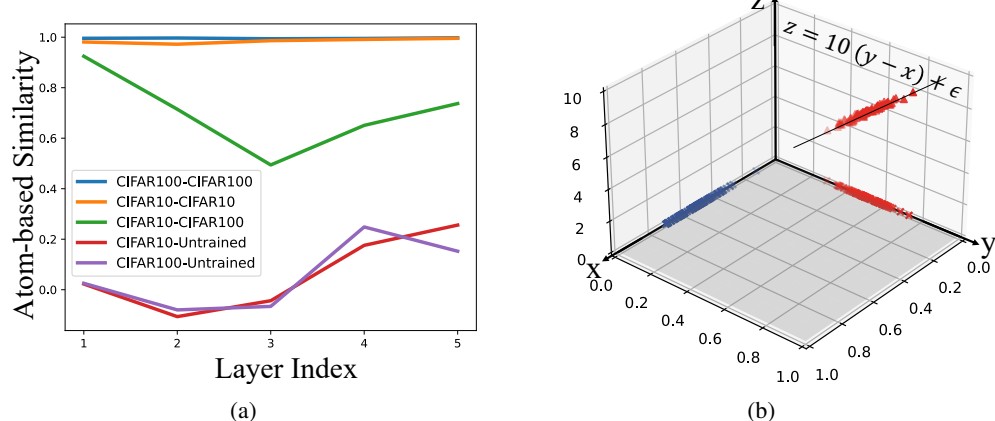

(a)                      (b)

Figure 9: (a) Using atom-based similarity, models trained on different datasets (CIFAR-10 and CIFAR-100) are similar among themselves, but they differ from untrained models. (b) Illustration of limitations of stimulus-based similarities. Input data from "red" ($\{(x_i = 0, y_i)\}$) and "blue" ($\{(x'_i = y_i, y'_i = 0)\}$) are orthogonal. Since two models are learned on "red" data, their similarity should be 1, which can be faithfully indicated by our atom similarity. However, stimulus-based similarities will become 0 with the "blue" probing data.

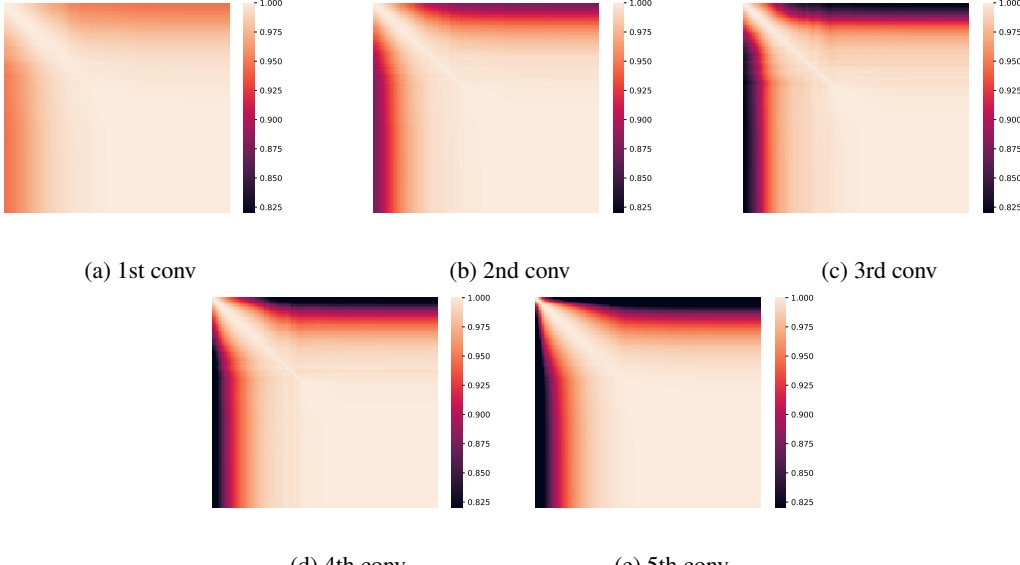

(a) 1st conv           (b) 2nd conv           (c) 3rd conv

(d) 4th conv           (e) 5th conv

Figure 10: Similarity of AlexNet with atoms from different time point during the training.

**Limitation of stimulus-based methods.** As shown in Figure 9(b), to illustrate sensitivity of stimulus-based similarities to probing data, we perform a simple regression task with data, $\{(x_i = 0, y_i, z_i)\}_{i=1}^n$, where $z_i = f(x_i, y_i) + \epsilon_i$ and $y_i, \epsilon_i \sim \mathcal{N}(0.5, 0.1)$. Two NN models $\mathcal{F}_1$ and $\mathcal{F}_2$ with the same initialization and atom coefficients are trained for their different atoms to learn $\mathcal{F} : (X, Y) \rightarrow Z$. It is can be simply found that the atom-based similarity of $\mathcal{F}_1$ and $\mathcal{F}_2$ is 1 and the stimulus-based similarity is also 1 with the same $\{(x_i = 0, y_i)\}$ as the probing data. However, if we choose $\{(x'_i = y_i, y'_i = 0)\}$ as the stimuli data, then the stimulus-based similarities directly become **0** as the data are now orthogonal to model parameters.

**Training dynamics.** We investigate the training dynamics of AlexNet (Krizhevsky et al., 2012) and VGG (Simonyan & Zisserman, 2014) separately on CIFAR-100 (Krizhevsky et al., 2009) and ImageNet (Russakovsky et al., 2015). The details of training dynamics of models with atoms from different time point during the training are shown in Figure 10 and Figure 11. Moreover, we examine the similarity between the two participated models shared the same initialization trained only with atoms on two different tasks. The results is shown in Figure 12 and Figure 13. The difference is less

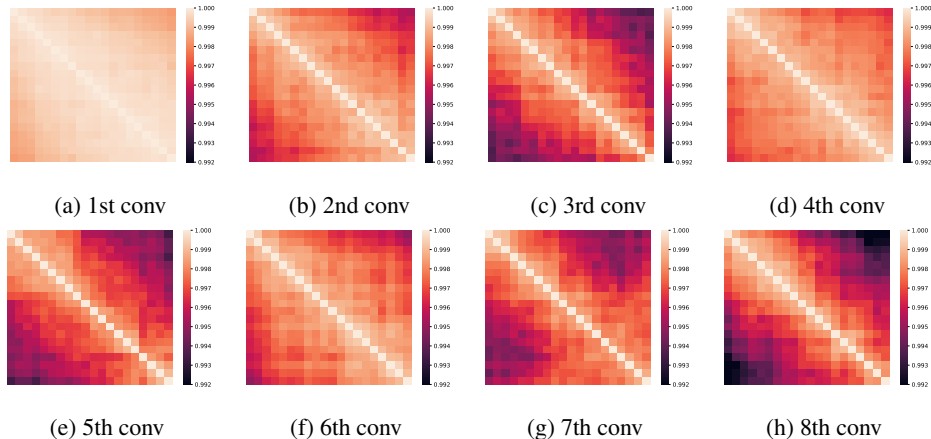

(a) 1st conv     (b) 2nd conv     (c) 3rd conv     (d) 4th conv

(e) 5th conv     (f) 6th conv     (g) 7th conv     (h) 8th conv

Figure 11: Similarity of VGG with atoms from different time point during the training.

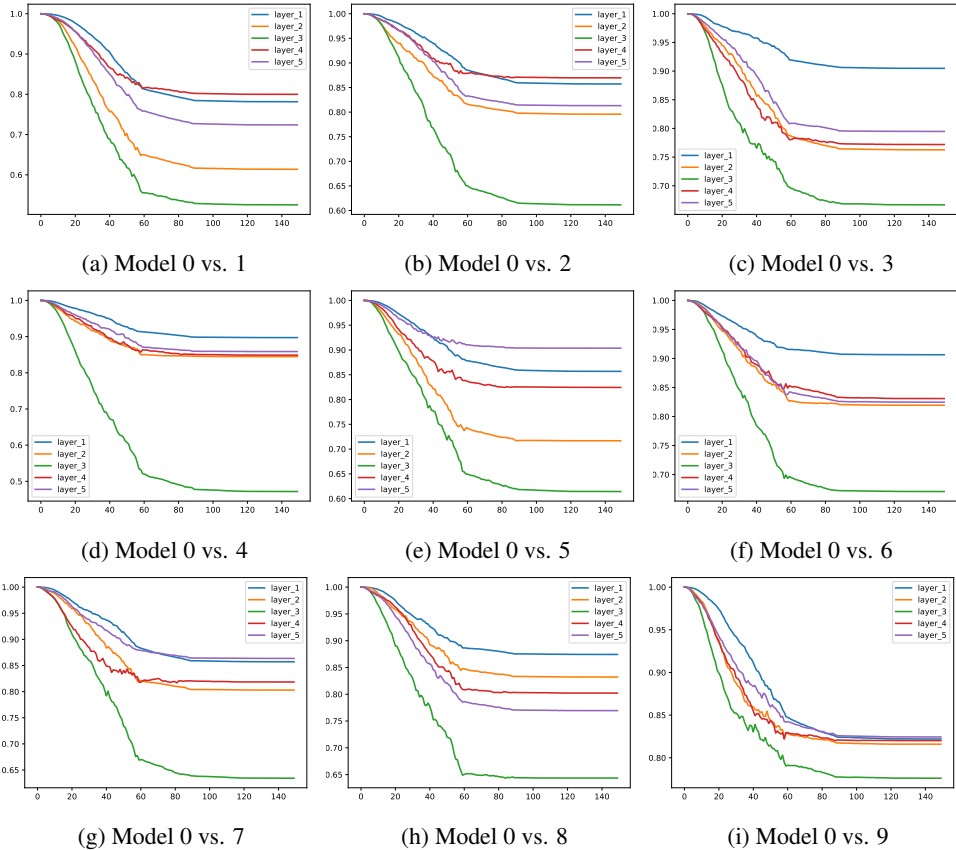

(a) Model 0 vs. 1     (b) Model 0 vs. 2     (c) Model 0 vs. 3

(d) Model 0 vs. 4     (e) Model 0 vs. 5     (f) Model 0 vs. 6

(g) Model 0 vs. 7     (h) Model 0 vs. 8     (i) Model 0 vs. 9

Figure 12: Similarity of AlexNet trained on different tasks during the training.

on the first few layers, but more on the middle layers. It reflects the middle layer is more critical than other layers, which is aligned with previous work (Neyshabur et al., 2020).

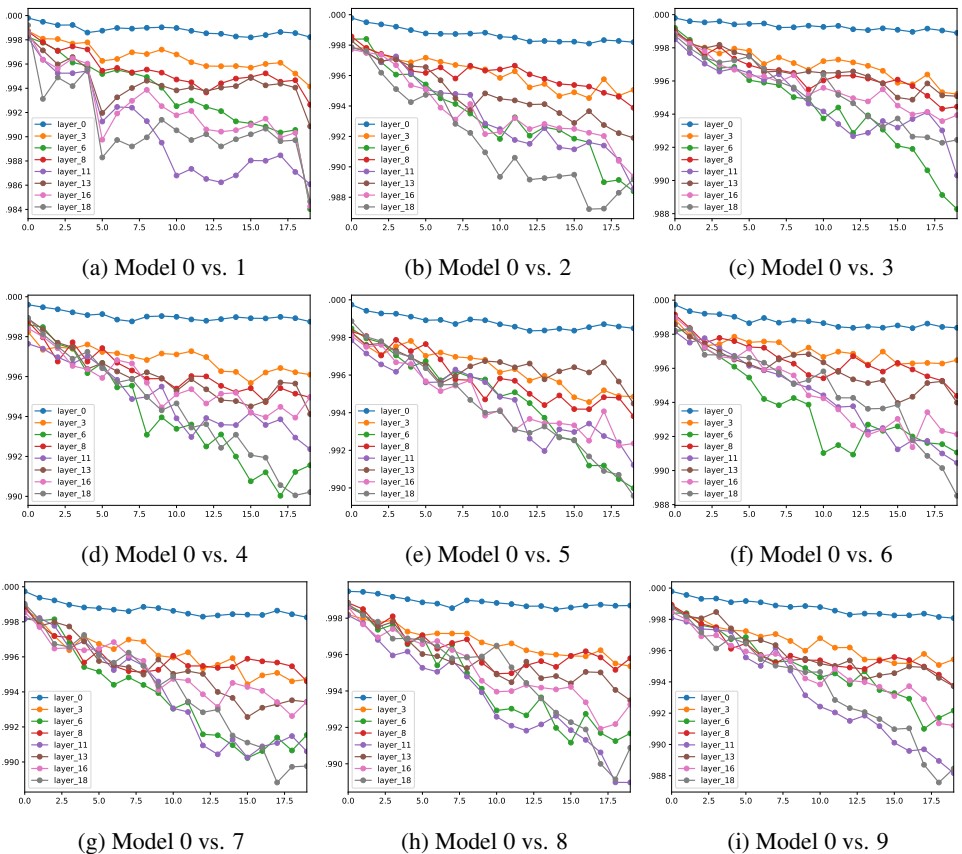

Figure 13: Similarity of VGG trained on different tasks during the training.

