# OpenReview forum: "Exploring Neural Network Representational Similarity using Filter Subspaces"
_ICLR.cc/2023/Conference — Submitted to ICLR 2023_

### Official Review · Reviewer_Sx1E · 2022-10-24

**Confidence:** 3
**Correctness:** 3
**Technical Novelty And Significance:** 3
**Empirical Novelty And Significance:** 3
**Recommendation:** 6

**Clarity, Quality, Novelty And Reproducibility:**

Please refer to the review section.


**Strength And Weaknesses:**

Please refer to the review section.


**Summary Of The Paper:**

This paper proposes a new paradigm for reducing representational similarity analysis in CNNs to filter subspace distance assessment. Model representational similarity can be significantly simplified when filter atom coefficients are shared across networks by calculating the cosine distance among respective filter atoms. This study demonstrates that this simplified filter subspace-based similarity preserves a strong linear correlation with other popular stimulus-based metrics while being significantly more efficient and robust. Overall this is an interesting paper. Major concerns and minor comments are presented in the review section.

**Summary Of The Review:**

a) The primary concept that is unclear to me is that the paper claims that the proposed method is linear. How stacked layers of ConvNets can be considered linear. I believe that it should be discussed with some mathematical evidence.

b) The performance of the proposed method should be compared with state-of-the-art representational learning approaches, such as Deep CCA (Andrew 2013), Deep GPs (Damianou 2013), Representational Similarity Learning (Oswal 2016), and Energy-Based Processes for Exchangeable Data (Yang 2020).

c) Lemmas 1 and 5 did not follow the same format to denote matrices – i.e., they should be highlighted in bold.

d) There are some minor linguistic and typo problems in this paper.

---

> ### Author Response · Authors · 2022-11-13
> **Response to Reviewer Sx1E**
>
> Thank you for your supportive comments.
>
> In this paper, we consider assessing model similarities in scenarios where numerous models exist and relations among models are important. We show both theoretically and empirically that the proposed filter atom-based similarity preserves a strong linear correlation with other popular stimulus-dependent similarity measures such as CCA. We demonstrate that our approach is highly efficient and immune to inappropriate choices of probing data.
>
> We hope the responses below will alleviate your concerns.
>
> **1. The primary concept that is unclear to me is that the paper claims that the proposed method is linear. How stacked layers of ConvNets can be considered linear. I believe that it should be discussed with some mathematical evidence.**
>
> As mentioned above, we claim that the proposed filter atom-based similarity and stimulus-based methods maintain a high correlation, which is theoretically analyzed with a single convolutional layer. When there exist stacked convolutional layers, we empirically demonstrate that, by varying the condition in assumption 1, the correlation between CCA and atom-based similarity keeps increasing to near linear.
>
> **2. The performance of the proposed method should be compared with state-of-the-art representational learning approaches, such as Deep CCA (Andrew 2013), Deep GPs (Damianou 2013), Representational Similarity Learning (Oswal 2016), and Energy-Based Processes for Exchangeable Data (Yang 2020).**
>
> These papers provide very insightful discussions about representational learning, which can provide a new venue for utilizing our method.
>
> DCCA (Andrew 2013) learns to generate correlated representations of two views of data by complex nonlinear transformations. Deep GPs (Damianou 2013) introduces a framework for efficient Bayesian training of hierarchical Gaussian process mappings. Representational Similarity Learning (Oswal 2016) aims to discover important features to locate the similar network structure of the brain. Energy-Based Processes for Exchangeable Data (Yang 2020) deal with problems that involve conditional and unconditional set distribution modeling.
>
> The methods provide in these papers are not directly applicable to measuring the model similarity. In our work, we compare our method with recent works, CCA (2017) and CKA (2019), which are two commonly used methods in representational similarity analysis.
>
>
> **3. Lemmas 1 and 5 did not follow the same format to denote matrices – i.e., they should be highlighted in bold.**
>
> We have incorporated that in the revised manuscript.
>
> **4. There are some minor linguistic and typo problems in this paper.**
>
> We have checked these issues and fixed that in the revised manuscript.

---

### Official Review · Reviewer_kifh · 2022-10-25

**Confidence:** 2
**Correctness:** 3
**Technical Novelty And Significance:** 2
**Empirical Novelty And Significance:** 2
**Recommendation:** 6

**Clarity, Quality, Novelty And Reproducibility:**

I encountered several unclear passages that I hope the authors can clear up for me:

* Why are the filter atoms first indexed by superscripts $\mathbf{D}^m$ and elsewhere as subscripts $\mathbf{D}_u$?

* Is the equality in Theorem 2 actually an equality? Or only an equality when $\mathbf{Z}_u^\top \mathbf{Z}_u$ is strictly diagonal?

* In section 2.1 the authors define $\mathbf{Z}$ to be post-nonlinearity activations, but they seem to define $\mathbf{Z}$ as pre-nonlinearity activations in section 2.4

* In section 2.4 the authors write $\mathbf{Z}_u = \boldsymbol{\alpha} \mathbf{X}_p \mathbf{D}_u$, but I don't know what this means since all three of these symbols are higher-order tensors and not matrices?

* The federated learning experiment (section 3.3) doesn't provide a lot of detail about how shared atom coefficients are achieved in practice. These details are very important to interpret the results and application and shouldn't be left to the Appendix. Similarly the details in the continual learning application (section 3.4) are pretty thin.

* Section 4.2 seems mostly superfluous and unnecessary. I suggest cutting this and adding more details in section 3.3 and 3.4

* One final comment/suggestion. The authors may consider remarking that their similarity measure becomes a proper metric if one takes the arc-cosine of the filter similarity. Several papers have recently argued that having a formal metric space over representations is important (see e.g. https://arxiv.org/abs/2110.14739)

**Strength And Weaknesses:**

This paper has a relatively detailed theory section and a clever idea. I think it is pretty interesting that weight space similarity and representational similarity can be correlated.

However, there are several weaknesses which have muted my enthusiasm.

* It is not clear to me that the computational expense of CKA (and related metrics) is a major bottleneck to research. So while the paper presents a potentially useful way to speed up calculations, I am not sure this enables fundamentally new research directions.

* The authors seem to claim that a weakness of CKA is that it is stimulus-dependent. However, I would claim the opposite &mdash; it is potentially very useful to compare CKA values on in-sample vs out-of-distribution inputs to understand how hidden layer representations are sensitve to these shifts. By focusing only on weight space similarity, the authors' approach is less flexible.

* The authors' framework requires the atom coefficients, denoted $\alpha$, to be shared across different networks. This means that the authors' method is less generally applicable than CKA. For example, it would be useful to compare representational similarity between the DCFNet architecture (which includes the atom coefficients) and more standard architectures. However, while it would be possible to make this comparison with standard CKA, there isn't a way to do this with the authors' method.

* While there is a lot of theoretical analysis in this paper, certain technical portions of the manuscript are not clear (see below).

* Related to above, I think that certain technical assumptions are not made explicitly. For example, it would seem to me that the atom coefficients need to be full-rank in order for the filter subspace similarity to be meaningful. Consider if $\alpha_{ij} = 1$, then any permutation of the filter atoms, $\mathbf{D}^1, \dots, \mathbf{D}^m$, would produce the same convolutional filter $\mathbf{W}$. Is my understanding here correct? If so, the conditions under which the filter atoms are directly comparable across networks needs further unpacking.

**Summary Of The Paper:**

The authors propose a new metric to evaluate representational similarity in deep neural networks. They aim to capture the properties of the well-known centered kernel alignment (CKA; Kornblith et al., 2019), but with substantially faster computation times. To do this, they leverage work by Miao et al. (2021) in which a group of neural networks is trained on different tasks but with a shared "atom coefficients" that render their weights directly comparable.

**Summary Of The Review:**

Overall, I think this work is potentially interesting. I would have liked to have seen it focused on when and why weight space similarity coincides with representational similarity in deep nets. Instead, the authors sold this new method as being a computationally cheaper and stimulus-independent form of CKA. I am less enthusiastic about this conclusion. The notation and clarity of the manuscript could also be improved. For now, I don't think this is quite ready and am categorizing this as a borderline reject, although I might be persuaded to increase my score if others find this interesting and if the text/equations can be clarified as mentioned above.

---

> ### Author Response · Authors · 2022-11-13
> **Response to Reviewer kifh (Part 1)**
>
> Thanks for your constructive comments.
>
> In the paper, we demonstrate the proposed method in the setting of federated learning (FL) and continual learning (CL), where numerous models exist, relations among models are usually critical, and the computational cost to assess the model relation can be a major bottleneck.
>
> As discussed in Section 3.3, the FL setting usually contains multiple clients. For example, these clients could be cellphones, where a recommendation system is performed based on users’ historical behaviors [1]. By utilizing model similarity metrics, the system can keep improving the recommendation performance of each client with support from similar ones, e.g., by simply conducting model ensemble. Our experiments only consider 100 clients, yet the speed advantage between our method and alternative stimulus methods is already remarkable. In the case of 10 thousand clients, based on the results in Table 1, the estimated time of calculating model similarity with our filter atom-based method is **2.6** minutes, while CKA will take about **10.7** days.
>
> **1. Usefulness of computation speed up**
>
> As mentioned above, our work focuses on application scenarios where numerous models exist, such as federated learning and continual learning. The computational cost may be neglectable while comparing the two models, but to calculate the similarity, e.g., among a large number of clients in the FL setting, the popular stimulus-based methods are significantly slower than the proposed filter atom-based approach and potentially become inapplicable.
>
> **2. Focusing only on weight space similarity loses flexibility in understanding in-sample vs out-of-distribution inputs.**
>
> We agree with the reviewer that CKA can be useful in OOD detection. However, what we mainly focus in the paper is the robustness in measuring the model similarity. For example, in the table of Figure 4, the in-sample and out-of-distribution data lead to inconsistent model similarity assessment with stimulus-based methods, while the similarity result of the proposed filter atom-based method aligns with the expectation.
>
> **3. Shared atom coefficients across different networks are less generally applicable than CKA.**
>
> In this paper, we consider assessing model similarities in scenarios where we can train and access numerous models. Federated learning and continual learning are two typical yet critical application scenarios in our considerations, as explained in the Introduction and Experiments sections. In those application scenarios, the proposed method is significantly more efficient and robust than CKA. The recent literature has demonstrated a paradigm with decomposed filter atoms and atom coefficients $W = \alpha \times D$ reporting superior performance than other state-of-the-art approaches, e.g., in continual learning [2]. We provide a highly efficient way to assess model similarity for this new paradigm, with detailed theoretical and empirical analysis.
>
>
> **4. Need to assume the atom coefficients to be full-rank.**
>
> Yes, atom coefficients should be full-rank. We have explicitly stated this assumption in the revised manuscript. The dimension of the atom coefficient is $c_{in} \cdot c_{out} \times m$, where $m$ is usually around 10, but $c_{in} \cdot c_{out}$ is usually over 100. In practice, even without any regularization, the full row-rank of atom coefficients is satisfied in almost all cases.
>
> **5: Why are the filter atoms first indexed by superscripts Dm and elsewhere as subscripts Du?**
>
> In Section 2.1, the superscript $D^i$ denotes the $i$-th atom, but subscript $D_u$ denotes the $m$ filter atoms. We have changed $D^i$ into $D[i]$ in the revised manuscript.
>
> **6: Is the equality in Theorem 2 actually an equality? Or only an equality when $Z_u^{\intercal} Z_u$ is strictly diagonal?**
>
> Yes, it is an equality. In the discussion after Theorem 2, we state that $\gamma_1$, $\gamma_2$ and $\gamma_3$ contain higher order of features, which leads to a non-linear correlation between CCA and atom-based similarity. As $Z_u^{\intercal} Z_u$ approaches the diagonal, the higher-order items will be neglectable, and the correlation will be linear, which is shown in Figure 4(b).
>
> **7. In section 2.1 the authors define $Z$ to be post-nonlinearity activations, but they seem to define $Z$ as pre-nonlinearity activations in section 2.4**
>
> To simplify the analysis, we assume that the post-nonlinearity activations $Z=\sigma(\alpha X D)$ have equivalent pre-nonlinearity activations $Z’=\alpha X’ D$ with certain choice of input $X’$. Specifically, we can write $Z=\sigma(\alpha X D) = \alpha X D + Z’’ =\alpha X’ D$ where $Z’’$ contains the non-linear elements. There exists $X’’$ which satisfies $Z’’=\alpha X’’ D$, so that $X’ = X + X’’$. As shown in Theorem 1 and 2, the $X$ degenerates into constant terms, which thus does not affect our conclusion.

---

> > ### Author Response · Authors · 2022-11-13
> > **Response to Reviewer kifh (Part 2)**
> >
> > **8. In section 2.4 the authors write $Z_u=\alpha X_p D_u$, but I don't know what this means since all three of these symbols are higher-order tensors and not matrices?**
> >
> > We use simplified notation to provide a basic idea of our formulation. We will add the details to the paper for clarification. Specifically, the formulation writes as $Z_u = \alpha X_p \star D_u$, where $\star$ is the convolutional operation. By converting convolutional kernel $D_u$ into a Toeplitz matrix, we can replace the convolution operation $X_p \star D_u$ with matrix multiplication $X_p D_u$.  We also modify $\alpha$ by $I_{hw} \bigotimes \alpha$, where $\bigotimes$ is Kronecker product, to enable the matrix multiplication $\alpha X_p D_u$.
> >
> > **9. More experiment details of FL and CL. Section 4.2 seems mostly superfluous and unnecessary. I suggest cutting this and adding more details in section 3.3 and 3.4**
> >
> > Section 4.2 provides a background of the current machine learning paradigms with numerous models. We have made this section more compact and added more experiment details in Section 3 in the revised manuscript.
> >
> > **10. proper metric**
> >
> > This is a good suggestion. We will add this as a remark.
> >
> > [1] ​​Li, T., Sahu, A. K., Talwalkar, A., & Smith, V. (2020). Federated learning: Challenges, methods, and future directions. IEEE Signal Processing Magazine, 37(3), 50-60.
> >
> > [2] Miao, Z., Wang, Z., Chen, W., & Qiu, Q. (2021). Continual Learning with Filter Atom Swapping. In International Conference on Learning Representations

---

> > > ### Comment · Reviewer_kifh · 2022-11-25
> > > **Still borderline... But going to raise to borderline accept**
> > >
> > > I thank the authors for their detailed responses and for clarifying most of my notational questions. I think they authors should add additional clarification about how $\gamma_1$, $\gamma_2$, and $\gamma_3$ "contain higher order of features". Can you please be explicit about what these three terms are a function of?
> > >
> > > I am still pretty conflicted about this paper and agonized over whether I should raise my score. My biggest remaining problem is that the motivation in the federated learning setting is not made clear in the abstract and introduction. I think making that change would greatly improve the quality and impact of the paper. The introduction should say something like -- "CKA is important for federated learning because it enables model ensembling [ADD CITATION]... However, this is too computationally challenging in practice [ADD CITATION OR BACK-OF-THE-ENVELOPE CALCULATION]... Thus, we developed a computationally efficient approach that approximates CKA..."
> > >
> > > I am still not 100% convinced that there is a strong motivating application for this approach, but federated learning is not my specialty so I will give the authors the benefit of the doubt. I am raising my score to borderline accept and decreasing my confidence score from 3 to 2.

---

> > > > ### Author Response · Authors · 2022-11-29
> > > > **Thank you for the response**
> > > >
> > > > Thanks for your constructive comments.
> > > >
> > > > Q1. Additional clarification about $\gamma_1$, $\gamma_2$, and $\gamma_3$
> > > >
> > > > As we discussed in Appendix, $\gamma_1=\kappa_F((Z_u^{\intercal}Z_u)^\frac{1}{2}) \cdot \kappa_F((Z_v^{\intercal}Z_v)^\frac{1}{2})$, where $\kappa_F(\cdot)$ is the condition number.
> > > > $\gamma_2$ and $\gamma_3$ are approximately 1. Specifically, we have $\gamma_2=\sqrt{1-\frac{Δ}{γ_1^2 γ_3^2} \frac{1}{cos^2(D_u, D_v)}}$, and since $Δ$ are small, with Taylor expansion, $\gamma_2 \approx 1-\frac{1}{2}\frac{Δ}{γ_1^2 γ_3^2} \frac{1}{cos^2(D_u, D_v)}$. The term $\frac{1}{cos^2(D_u, D_v)}$ causes non-linearity in the relation between CCA and atom-based similarity. We have added the explanation in the revised manuscript.
> > > >
> > > > Q2. Concern about the motivation in federated learning is not made clear in the abstract and introduction
> > > >
> > > > In this paper, we focus on assessing model similarities in numerous-model scenarios, where federated learning and continual learning are two cases.
> > > > Based on the advice, to make our message clearer, we have incorporated the suggestion in the Introduction.

---

### Official Review · Reviewer_KvFf · 2022-10-30

**Confidence:** 3
**Correctness:** 3
**Technical Novelty And Significance:** 2
**Empirical Novelty And Significance:** 2
**Recommendation:** 3

**Clarity, Quality, Novelty And Reproducibility:**

The premise of the model is not well explained and assumptions are stated early on are not clear.

**Strength And Weaknesses:**

Strengths:
+ Validated their approach in diverse contexts, continual learning and federated learning setting where different distributed learners have access to different batches of data and are learning locally.

+ Major benefits in terms of savings if filter subspaces are analyzed rather than the representations directly.

+ They show that on simple tasks, analyzing filter subspaces is very correlated with the RSA-based approach

Weaknesses:

- There appear to be fundamental differences in what representational similarity is after vs. what this method provides. Thus it’s unclear if the claims of “the proposed atom-based method can achieve millions of times computation reduction than popular stimulus-based methods” is entirely fair since the methods aren’t trained/tested under the same assumptions (where RSA is a black-box method).

The assumptions are unclear:

- The authors start with the assumption that “two convolution neural networks Fu, Fv share atom coefficients layer-wise”.
When is this assumption satisfied? Do you need to train networks in a particular way to align them or to ensure that this constraint holds?

- Because similarity is implicit, it seems that for the comparisons in the first propositions to hold, you need assume that the networks are trained on the same data streams. Either way it seems that the bounds linking the filter subspaces and representational similarity would be very loose.

- Does this only work on CNNs? Why make this assumption?


**Summary Of The Paper:**

This paper explores the use of the filter subspace distance as a measure of similarity between two neural networks. They provide theoretical and empirical justifications for this measure of similarity and validate their atom-based similarity on continual learning and federated learning tasks.

Most measures of representational similarity (like CCA and CKA) require a forward pass through a trained model to compute representations of different inputs at different layers in a model. This paper takes a different approach and aims to decompose the filter weights of the network directly and then compare the underlying subspaces that are spanned by the network weights. By avoiding having to compute the representations directly, they can avoid the computational overhead needed in other existing approaches.

While it is likely that the weights in two networks, when tuned or aligned, can help to reveal differences between the two, the claims in the abstract that they can achieve million times reduction in computation appears overstated because the setup for both approaches and goals is very different -- as RSA and the other tested approaches are black-box. There are other approaches that use weight distributions to compute measures of similarity and perhaps would be more appropriate to compare here.

Overall, the assumptions underlying their framework aren’t very well described and it's unclear how and when their metric will provide appropriate measures of similarity for different pre-trained models.


**Summary Of The Review:**

This paper provides new tools for measuring similarity between networks using their weights rather than representations. While some of their results are promising and show that their approach can be helpful in continual and federated learning settings, none of the baselines are weight-based analysis approaches, the assumptions underlying their framework aren’t very well described, and it's unclear how and when the atom-based metric will provide appropriate measures of similarity for different pre-trained models.

---

> ### Author Response · Authors · 2022-11-13
> **Response to Reviewer KvFf**
>
> Thanks for your constructive comments. We address all your suggestions in the following and hope the responses will alleviate your concerns.
>
> **1. A fair comparison with the RSA method**
>
> In this paper, we focus on assessing model similarities in scenarios where we can train and access numerous models with the same architecture. Federated learning and continual learning are two typical yet critical application scenarios in our consideration, as discussed in the Introduction and Experiments sections. In these cases, our method and stimulus-based approaches can both be adopted for model similarity assessment. As the same setup and goal are assumed here, we consider it a fair comparison between our method and stimulus-based approaches.
>
> In aforementioned application scenarios, due to the ‘black-box’ nature, stimulus-based methods can experience high variances in similarity measures while using different probing data. As shown in the table in Figure 4, two models trained on the same dataset (CIFAR100) are supposed to have a high similarity, but stimulus-based metrics report completely different results for different probing data (CIFAR100 and SVHN). Moreover, in the setting of federated learning where sharing data is typically not allowed, the “black-box” methods can even be inapplicable. In contrast, our method captures intrinsic similarities between neural networks with the proposed filter atom metric, for more reliable and useful model assessment in the aforementioned applications. This is also illustrated in the table of Figure 4 that our method reliably captures the high similarity between two models learned from the same dataset.
>
> To the best of our knowledge, we cannot find additional weight-based model similarity measure to compare with the proposed method. We will make a comparison if the reviewer may kindly suggest any additional references.
>
> **2. How to get the shared atom coefficients?**
>
> As discussed in the paper, our framework enforces a convolutional filter $W$ to be decomposed as filter atoms $D$ and atom coefficients $ \alpha$, *i.e.*, $W = \alpha \times D$. With this formulation, we consider a paradigm where atom coefficients are shared across different deep models while only filter subspaces are model-specific. This paradigm has recently reported in the literature the state-of-the-art performance in scenarios such as continual learning [1], where each model shares the same atom coefficients and only learns its own filter atoms on the corresponding task. This paradigm, where numerous neural networks are typically learned, enable us a completely new way to assess the model similarity based on filter atom similarity.
>
> For example, as shown in Section 3.3, different clients in FL share the same atom coefficients, but keep training the filter atoms on their individual client data. The shared atom coefficients can be obtained by training on a subset of tasks, or some pre-training tasks, e.g., ImageNet, etc. Since only the filter atoms of CNNs are trained while atom coefficients are fixed, some implicit alignment is enforced across models, thus we can now directly compare the similarity between models without additional alignment, which is supported by our theoretical and empirical justification in the paper.
>
>
> **3. Bound from the proposition 1**
>
> We would like to clarify that Proposition 1 reveals the relationship between the change in output and atoms in a decomposed convolutional layer, which does not depend on the training. To be specific, the data $X$ in the proposition is the probing data to generate features of two convolutional layers, which share the same atom coefficients but contain different filter atoms. Two filter atoms $D_u, D_v$ can be two random atoms and thus are not required to be trained on the same data. Thus, the first proposition can upper bound corresponding features from any atoms without dependencies on the training process.
>
> **4. Does this only work on CNNs?**
>
> This work only focuses on CNNs, as the proposed method is mainly based on the findings in the filter subspace. Additional models other than CNN will be explored in our future work.
>
> [1] Miao, Z., Wang, Z., Chen, W., & Qiu, Q. (2021). Continual Learning with Filter Atom Swapping. In International Conference on Learning Representations

---

### Decision · Program_Chairs · 2023-01-20

**Decision:**

Reject

**Justification For Why Not Higher Score:**

Even if the core ideas are valid, the assumptions and setting under which it can be applied are not clear currently and major rewriting would be needed. Ultimately the question is how useful this paper will be to the readers in its current format: it is difficult for a practitioner to understand at which scenarios this method can be employed, and it is also difficult for a researcher to build on top of this work without all the assumptions being investigated more thoroughly.

**Justification For Why Not Lower Score:**

N/A

**Metareview: Summary, Strengths And Weaknesses:**

This paper considers the problem of estimating the similarities among the representations of different neural networks. The authors are motivated by results of prior work which uses filter subspaces for continual learning, and use it to solve the similarity problem. In turn, this solution can be used within a continual learning and federated learning task. A key goal for this work is to obtain a quick to estimate similarity score (because the method does not require to compute representations of different inputs). The authors also provide theoretical analysis which the reviewers found correct.

Overall the reviewers found the main ideas valid and interesting and also appreciated the theoretical analysis provided. The general research area of measuring some sort of similarity among networks is also an interesting one, and linking that to continual learning and federated learning is an interesting direction.

However the reviewers also have several concerns, even after the rebuttal and the discussion period (including the discussion that has happened among reviewers privately). Firstly, the word "similarity" is overloaded, and the reviewers have requested a clearer explanation of what "similarity" means here and under which assumptions. Furthermore, the method requires shared atom coefficients; the authors explain how these can be obtained especially motivated by continual and federated learning setups, however again it is not entirely clear how these assumptions potentially constrict the generality and applicability of the method compared to the other baselines. Adding to this, the analysis only works for CNNs.

Another issue with how the paper is currently motivated is that it is unclear at which scenarios the provided computational speedup is useful. The authors mention in the rebuttal the case of having to train many models for continual and federated learning, but the experiments are rather simplistic in the sense that they do not demonstrate why in practice the provided speedup would make a difference in a real setting.

Overall it seems that this paper proposes valid ideas, but the assumptions and setting under which it can be applied are not clear in its current format.